# A Single Layer to Explain Them All:
# Understanding Massive Activations in Large Language Models

**Zeru Shi**[1]   **Zhenting Wang**[1]   **Fan Yang**[2]   **Qifan Wang**[3]   **Ruixiang Tang**[1]

## Abstract

We investigate the origins of massive activations in large language models (LLMs) and identify a specific layer named the **Massive Emergence Layer (ME Layer)**, that is consistently observed across model families, where massive activations first emerge and subsequently propagate to deeper layers through residual connections. We show that, within the ME Layer both the RMSNorm and the FFN parameters jointly contribute to the emergence of massive activations. Once formed, the massive activation token representation remains largely invariant across layers, reducing the diversity of hidden representations passed to the attention module. Motivated by this limitation, we propose a simple and effective method to reduce the rigidity of the massive activation token. Our approach consistently improves LLM performance across multiple tasks, including instruction following and math reasoning, in both training free and fine tuning settings. Moreover, we show that our method mitigates attention sinks by selectively weakening their influence, elucidating their origin at the hidden state level and shedding new light on principled mitigation strategies. The model and code have been released at MELayer & WeMask.

## 1. Introduction

Large Language Models (LLMs) (Yang et al., 2025; Liu et al., 2024) have demonstrated strong capabilities across a wide range of complex tasks, motivating increasing efforts to probe their internal mechanisms (Zhao et al., 2024; Shi et al., 2025; Zhang et al., 2025c;b). Some work use embeddings to following tasks (Shi et al., 2026). One emerging line of work focuses on **massive activations**: in intermediate representations, the embeddings of few tokens can attain values several orders of magnitude larger than the rest. This raises a fundamental question: *why do such extreme activations arise in LLMs, what do they encode, and how do they shape model behavior?* Recent studies suggest that massive activations can behave like dominant bias terms (Sun et al., 2024), affect contextual information processing (Jin et al., 2025), and alter attention behavior and training dynamics (Kaul et al.; Gallego-Feliciano et al., 2025). Despite these advances, existing work still lacks a clear account of how massive activations emerge end-to-end and how their emergence connects to their downstream functional effects in LLMs.

In this paper, we provide a systematic analysis of the emergence of massive activations in LLMs. We find that massive activations are generated at a **single layer** of the model and, once formed, propagate to subsequent layers through residual connections. As shown in Figure 1 and Appendix H, in the particular layer, the activation values of the massive activation tokens will increase by several hundreds times compared to the previous layer. We refer to this layer as the **ME Layer** (*M*assive *E*mergence Layer). In Figure 1, we illustrate how massive activations are generated at the ME Layer and then propagate into later layers. Surprisingly, we show that the *ME Layer is consistently observed across models of different sizes and families* (see Appendix H), suggesting a shared, architecture-level mechanism and positioning the ME Layer as the primary locus for systematic analysis of massive activation emergence.

To unpack the ME Layer mechanism, we conduct a fine-grained analysis within this layer and find massive activation emergence is jointly driven by the pre-FFN RMSNorm and the FFN layer in the ME Layer. We further find that massive activations exhibit high degree of stability and consistency (subsection 3.2 and Appendix D). This invariance reduces representation diversity. When it propagates into self-attention, the shared direction biases how tokens interact, making attention patterns more similar across inputs and less context-adaptive in practice.

To mitigate the effects of massive activation–induced directional invariance in hidden states, we propose a method that

*Equal contribution  [1]Rutgers University  [2]Wake Forest University  [3]Meta AI. Correspondence to: Ruixiang Tang <ruixiang.tang@rutgers.edu>.

*Proceedings of the 43rd International Conference on Machine Learning*, Seoul, South Korea. PMLR 306, 2026. Copyright 2026 by the author(s).

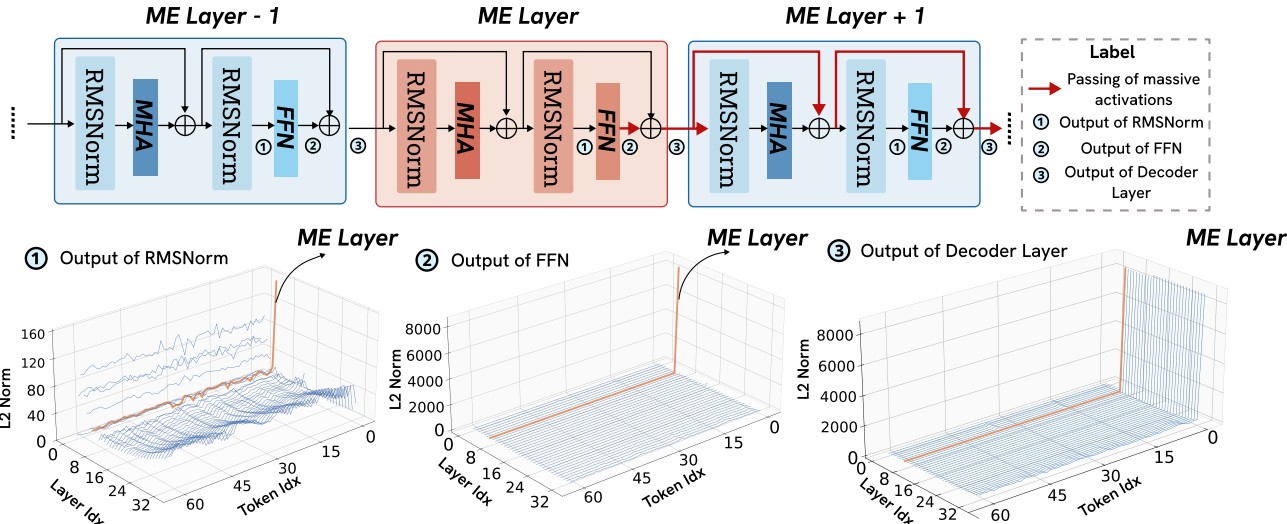

*Figure 1.* This figure illustrates how massive activations emerge and propagate. In the top panel, we trace the flow of massive activations: they arise only at the FFN of a specific layer and then propagate to subsequent layers through residual connections. The → arrows denote the generation and propagation of massive activations. The bottom panel shows how the output $\ell_2$ norm changes across layers. ME Layer means Massive Emergence Layer.

starts from the ME Layer and selectively masks dimensions in the attention input corresponding to large RMSNorm weights, which tend to amplify dominant directions in the hidden state. This operation relaxes the directional rigidity of the massive activation token while preserving the overall structure of the representation, thereby restoring greater directional diversity in the attention input. As a result, the attention mechanism can better adjust its similarity structure across different inputs. Experimental results show that our method consistently improves model performance across downstream tasks, both as an inference-time, training-free intervention and when applied during fine-tuning.

We further analyze the attention sink phenomenon (Xiao et al., 2024), in which LLMs assign disproportionately large attention weights to a small subset of tokens, typically the first token. We find that attention sinks emerge in the layer immediately following the ME Layer, and that the corresponding attention weights exhibit low-rank properties similar to those of the massive activations produced in the ME Layer. Our method leads to a partial attenuation of attention sinks, and that this controlled reduction is consistently associated with improved model performance. These results suggest a new perspective on attention sinks from a representational standpoint: attention sinks are not inherently detrimental, but instead appear to play a functional role in model computation. Rather than eliminating them entirely, moderately reducing their dominance while preserving their presence yields more effective and stable behavior, highlighting the importance of balancing representational flexibility with structural regularization.

In summary, our contributions are as follow:

- We trace the massive activation phenomenon back to its root cause and find **ME Layer**, the massive activation of hidden state starting from the this layer and propagate via residual connections.
- We show that massive activations arise from the characteristics of the RMSNorm and FFN weights in ME Layer, and the properties of the massive activation token remain highly consistent across different inputs and layers.
- We propose a method that relaxes the directional rigidity of the massive-activation token, enabling self-attention to respond more contextually across inputs and delivering consistent performance gains across multiple model families and tasks.
- We provide a new perspective on the attention sink phenomenon based on our findings, offering a hidden state level explanation of its origin and new insights into mitigating the bad influence of attention sink.

## 2. Related Work

### 2.1. Massive Activation

Timkey & Van Schijndel (2021) first identified the phenomenon that certain feature dimensions exhibit extremely large activations in GPT-2. Following this observation, several studies began to investigate such outlier features in hidden states (Dettmers et al., 2022; Zeng et al., 2022; Ahmadian et al., 2023). Subsequent work explored these outlier features from different perspectives: Owen et al. (2025) studied them through quantification analysis, while Zhao et al. (2025) examined their functional roles. Other studies attempted to suppress or remove outlier dimensions to improve model robustness or quantization (Bondarenko et al.,

2023). More recent work reported the presence of unusually large magnitude hidden states, often referred to as massive activations (Sun et al., 2024; Son et al., 2024). Oh et al. (2025) further suggested that such massive activations can be driven by large FFN weights. In addition, Gallego-Feliciano et al. (2025) analyzed how massive activations emerge during training, while He et al. (2024) investigated how massive activations affect model performance and behavior. Meanwhile, other studies argue that attention sinks may serve functional roles rather than being purely pathological artifacts; for example, Ruscio et al. (2025) and Zhang et al. interpret attention sinks as structural anchors in the model. In (Cancedda, 2024) and (Ferrando & Voita, 2024), they report the BOS token residual stream write in a "dark subspace" and this stability across layers. (Queipo-de Llano et al., 2025) develops a unified theory showing that massive activations explain both attention sinks and compression valleys, and uses this to motivate a Mix–Compress–Refine view of depth-wise computation. Despite these advances, existing work still lacks a unified analysis that connects the emergence of massive activations with their downstream effects particularly attention sinks and leverages such source level understanding to develop targeted mitigation methods.

## 2.2. Attention Sink

In LLM self-attention, a small subset of tokens consistently receives disproportionately large attention weights, a phenomenon known as attention sinks. Prior work observes attention sinks in both LLMs and VLMs (Xiao et al., 2024; Darcet et al.). Gu et al. (2024) characterizes sinks as non-informative key biases arising from softmax-induced coupling, motivating a line of work that mitigates sinks by modifying the attention mechanism (Ramapuram et al., 2024; Zuhri et al., 2025; Bondarenko et al., 2023; Miller, 2023). Representative approaches include attention gating and clipping (Bondarenko et al., 2023), gated attention modules (Qiu et al., 2025), and decoupling value states from sink dynamics (Bu et al., 2025). Some work also discuss the safety mechanism (Shang et al., 2025; Zhang et al., 2025a; Zhang & Zhang, 2025).However, existing analyses largely focus on attention, overlooking the role of embeddings.

# 3. Emergence of Massive Activations in a Single Transformer Layer

As shown in Figure 1, massive activations emerge abruptly within a single transformer layer, the ME Layer rather than accumulating gradually across layers. We analyze the origin of this phenomenon in subsection 3.1, linking it to the ME Layer 's normalization behavior and weight structure. In subsection 3.2, we further show that once formed, these activations become directionally stable, reducing representational diversity and constraining downstream self-attention.

## 3.1. Understanding the Emergence in the ME Layer

> **Key Takeaway**
>
> Massive activations emerge only at the ME Layer driven by unusually large and directionally aligned RMSNorm and FFN parameters that selectively amplify the massive-activation token.

In this section, we use Qwen3-4B as a case study to pinpoint the computations in the ME Layer that trigger massive activations. Figure 1 reveals a clear transition in activation magnitude centered at the ME Layer. Before this layer, token activations remain comparable across tokens, whereas at the ME Layer the first token exhibits a sudden and isolated increase in magnitude that is subsequently preserved through residual connections. The lower panels further localize this transition within the ME Layer : deviation first appears at the RMSNorm output and is sharply amplified by the FFN into a massive activation. Once formed, this large-magnitude representation is directly propagated to later layers. This staged behavior localizes the origin of massive activations to the internal transformations of the ME Layer. Among the components of a decoder block, only RMSNorm and the FFN can induce such rapid, token-specific amplification within a single layer, motivating a focused analysis of these two modules. We find that Qwen3-4B consistently exhibits massive activations on the first token across diverse inputs, accordingly, in the following sections, we use the first token as our primary object of analysis.

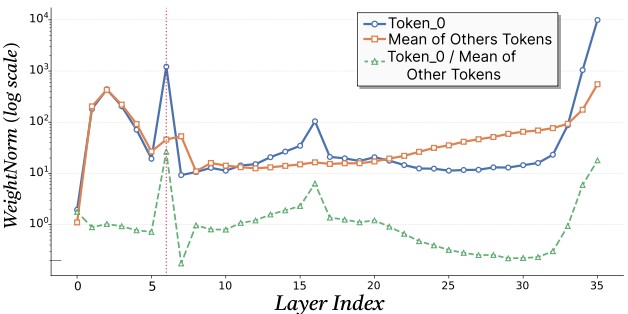

*Figure 2.* The comparison of the magnification of RMSNorm on $\text{token}_0$ and other tokens in Qwen3-4B across layers.

**Amplification effect of RMSNorm.** We analyze the scaling factors in RMSNorm layer by layer and find that the amplification effect in the ME Layer on the hidden state far exceeds that of other layers. In Figure 2, we measure the RMSNorm weighted activation norm, which represents the overall magnitude of the RMSNorm output for each token: $\text{WeightNorm}_l(t) = \left\| \hat{h}_{l,t} \right\|_2$, where $\hat{h}_{l,t} = \text{RMSNorm}(h_{l,t})$ denotes the output of RMSNorm at layer $l$ and token position $t$. We observe that before layer 7, the first token and the other tokens are amplified to a similar extent. However, at layer 7, RMSNorm produces a much

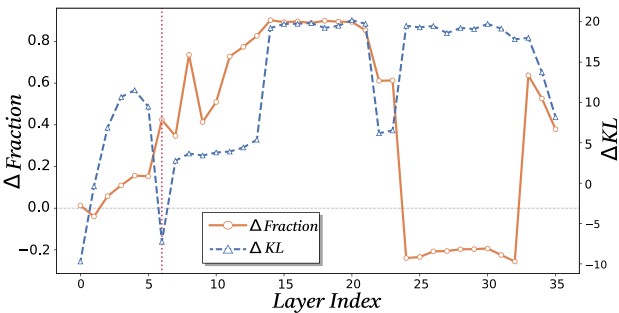

*Figure 3.* This metric captures the contribution of high-weight dimensions and reflects how well a token's values align with weight-based amplification across layers.

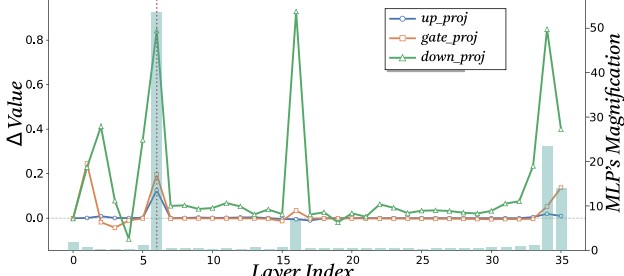

*Figure 4.* Line chart(the y-axis on the left) shows difference of the projection concentration between first token and others after different module in FFN. Bar chart(the y-axis on the right) shows the amplification factor of the MLP on the token hidden state.

larger output magnitude for the first token than for the other tokens. To further analyze whether this amplification is associated with dimensions corresponding to large RMSNorm scaling factors, we examine how the squared magnitude of the RMSNorm output is distributed across dimensions. Let $\mathcal{K}$ denote the index set of the top-$K$ largest RMSNorm scaling factors. We define the total squared magnitude of output as $E_t = \sum_{i=1}^{D} \hat{h}_{t,i}^2$, and the contribution from dimensions in $\mathcal{K}$ as $E_t^{\mathcal{K}} = \sum_{i \in \mathcal{K}} \hat{h}_{t,i}^2$. The fraction of the output magnitude contributed by high-scaling dimensions is then defined as $\text{Frac}_t = \frac{E_t^{\mathcal{K}}}{E_t}$. We compute the difference between the first token and the average of the remaining tokens as

$$\Delta\text{Frac} = \text{Frac}_0 - \frac{1}{S-1}\sum_{t=1}^{S-1}\text{Frac}_t. \quad (1)$$

Meanwhile, we also measure the similarity between the RMSNorm output distribution and the distribution induced by the RMSNorm scaling factors using KL divergence:

$$\Delta\text{KL} = \text{KL}(p_0 \,\|\, g) - \frac{1}{S-1}\sum_{t=1}^{S-1}\text{KL}(p_t \,\|\, g), \quad (2)$$

where $p_i = \frac{\hat{h}_i^2}{\sum_{j=1}^{D}\hat{h}_j^2}$, $g_i = \frac{f_i^2}{\sum_{j=1}^{D}f_j^2}$, and $f_i$ denotes the RMSNorm scaling factor of dimension $i$. As shown in Figure 4, at the ME Layer a large positive $\Delta\text{Frac}$ indicates that the RMSNorm output of the first token is more strongly concentrated on dimensions associated with large scaling factors, while a negative $\Delta\text{KL}$ shows that the overall output pattern of the first token is more consistent with the distribution induced by RMSNorm scaling. These results indicate that RMSNorm disproportionately amplifies the first token at the ME Layer through concentrated scaling effects.

**Amplification effect of FFN**   In addition to RMSNorm, the FFN in the ME Layer also contributes to the magnification of hidden states. To characterize how selectively a token's representation is shaped by the FFN, we compute the projection concentration, which measures how concentrated

the hidden state is along a small subset of representation dimensions after the FFN transformation. A higher projection concentration indicates that the resulting token representation is dominated by a limited number of projection induced directions, rather than being evenly distributed across the representation space. This metrics captures the downstream representational effect of selective activation induced by these projections. As such, projection concentration serves as an indirect indicator of how strongly the input representation is shaped by a small subset of FFN projection directions, rather than a uniform transformation across all dimensions. The formula is defined as follows:

$$\mathcal{C}_t = \sum_{i=1}^{d}\left(\frac{(h_{t,i})^2}{\sum_{j=1}^{d}(h_{t,j})^2}\right)^2, \quad (3)$$

$d$ denotes the hidden-state dimension, and $h_{t,i}$ denotes the $i$-th dimension of the $t$-th token. The results are shown in Figure 4. We observe that only at the ME Layer does the difference between the first token and the other tokens simultaneously reach its maximum across all three FFN modules. This indicates that, at the ME Layer, the first token exhibits a substantially stronger selective activation pattern under FFN transformations than in other layers, consistent with its disproportionately amplified activation at this layer. Meanwhile, we also report the amplification factor of the MLP for the first token. As shown in the figure, at the ME Layer the projection contributions of the three FFN projections jointly peak, resulting in the strongest amplification effect.

In Appendix B, we examine the respective contributions of RMSNorm and the FFN to the emergence of massive activations. The results highlight a complementary interaction between the FFN and the preceding RMSNorm within the ME Layer. Specifically, the FFN is the primary driver responsible for generating and sustaining massive activations, while the pre-FFN RMSNorm plays a critical role in regulating their scale. Together, these components amplify the massive-activation token to levels that are hundreds or even thousands of times greater than those of other tokens.

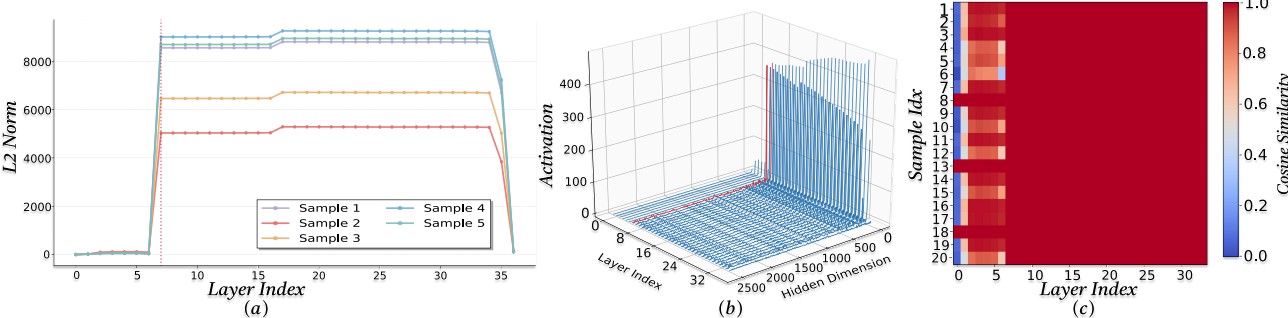

*Figure 5.* (a) L2 norm of the first token's hidden state across layers for different input instances. (b) The activation of token 0 in different layer of model. Red line indicates the activation of ME Layer (c) Heatmap of the cosine similarity between different input's first-token hidden state across layers.

## 3.2. The Direction of Massive Activation

> **Key Takeaway**
>
> Once the massive activation emerges at the ME Layer the massive activation's hidden state exhibits strong input-invariant directionality and remains stable across subsequent layers.

After identifying the ME Layer we further investigate the massive activation from the perspective of hidden states in the layers following ME Layer. We observe the value and direction of the hidden state of massive activation remain highly consistent across different tasks and input instances.

To identify the nature of the massive activation token, we similarly use Qwen3-4B as the representative model. Unlike models with an explicit begin of sequence token, Qwen3-4B does not introduce a dedicated start token embedding at the input. Therefore, any massive activation observed at a specific token position cannot be trivially attributed to a fixed or input independent embedding, but must emerge from the interaction between the input content and the model's internal transformations. We construct several different inputs from different tasks and compute: ❶ the L2 Norm of the massive activation's hidden state, ❷ massive activation token's hidden state across layers ❸ Cosine similarity of the massive-activation hidden states across layers with respect to a different input. The results are shown in Figure 5. As shown in Figure 5(a), once the massive activation emerges, the L2 norm of the massive activation remains stable across subsequent middle layers, indicating limited influence from later transformations. As shown in Figure 5(b), the hidden-state patterns of the massive activation remain similar across layers after the ME Layer suggesting that the activation direction is preserved. Consistently, Figure 5(c) shows that cosine similarity across different inputs remains nearly unchanged after the ME Layer. Therefore, it is well demonstrate that **the hidden state of the massive activation token remains stable across layers and inputs once it emerges**. More results in Appendix D and Appendix F.

## 4. Weight Guided Dimension Masking

Based on the previous analysis, we observe that after the ME Layer, the information encoded in massive activations remains largely identical across different inputs. While such massive activations can serve as a stable and shared global reference vector, a fixed hidden-state direction introduces inherent limitations. Once this direction becomes rigid, it restricts the attention mechanism's ability to conditionally adapt to diverse inputs, thereby reducing its input dependent flexibility during inference.

### 4.1. Directional Rigidity Constrains Attention

To understand why directional similarity persists when hidden states enter the attention module, we examine the effect of the pre-attention RMSNorm. Before attention, hidden states are normalized by RMSNorm, defined as $\mathrm{RMSNorm}(\mathbf{x}) = \frac{\mathbf{x}}{\sqrt{\frac{1}{d}\sum_{i=1}^{d} x_i^2 + \epsilon}} \odot w$, Without the learnable scaling vector $w$, RMSNorm strictly rescales the magnitude of the hidden state while preserving its direction. With learnable scaling, RMSNorm performs a dimension wise reweighting, which in general can alter the representation direction. However, in the regime we study, the massive activation's hidden state after the ME Layer exhibits highly concentrate along a small subset of dimensions. In such cases, dimension-wise scaling primarily amplifies already dominant components rather than introducing new directional components. As a result, although RMSNorm may change the exact direction, the dominant orientation of the representation remains largely consistent across inputs after normalization. Therefore, when entering the attention module, the massive activation's hidden state retains a highly similar direction across different inputs.

In self-attention, keys are obtained via a linear projection, $k_0 = h_0 W_K$. By decomposing the hidden state as $h_0 = \|h_0\|\hat{h}_0$, where $\hat{h}_0$ denotes the unit vector, we can rewrite the key as $k_0 = \|h_0\|(\hat{h}_0 W_K)$. This decomposition highlights that when the direction $\hat{h}0$ of the massive activation remains stable across inputs, the resulting key occupies

*Table 1.* This table reports the performance of our method across multiple benchmarks, evaluating the model's generalization ability after instruction fine-tuning. TF denotes a training-free inference-time setting without parameter updates, while SFT denotes supervised fine-tuning with parameter updates. **Bold** indicates the best performance under the corresponding experimental settings.

| Method | Mask Rate | MMLU ↑ | PIQA ↑ | ARC-C ↑ | MathQA ↑ | StrategyQA ↑ | Avg. |
|---|---|---|---|---|---|---|---|
| | | *Our Method(training free)* | | | | | |
| Qwen3-4B + SFT | - | $53.77 \pm 0.14$ | $80.30 \pm 0.00$ | $86.69 \pm 0.00$ | $37.25 \pm 0.00$ | $64.15 \pm 0.00$ | 64.43 |
| Qwen3-4B + SFT + WeMask(TF) | 0.1 | $\mathbf{54.32 \pm 0.17}$ | $80.69 \pm 0.00$ | $\mathbf{87.54 \pm 0.00}$ | $\mathbf{37.76 \pm 0.00}$ | $64.24 \pm 0.00$ | 64.91 |
| Qwen3-4B + SFT + WeMask(TF) | 0.3 | $54.23 \pm 0.23$ | $80.52 \pm 0.19$ | $87.26 \pm 0.35$ | $\mathbf{37.76 \pm 0.00}$ | $\mathbf{64.63 \pm 0.08}$ | 64.88 |
| Qwen3-4B + SFT + WeMask(TF) | 0.5 | $54.01 \pm 0.65$ | $\mathbf{81.03 \pm 0.22}$ | $86.92 \pm 0.05$ | $37.62 \pm 0.00$ | $64.19 \pm 0.08$ | 64.74 |
| Qwen3-4B + SFT + WeMask(TF) | 0.7 | $54.17 \pm 0.05$ | $81.01 \pm 0.00$ | $86.25 \pm 0.29$ | $37.49 \pm 0.00$ | $64.22 \pm 0.20$ | 64.63 |
| Qwen3-4B + SFT + WeMask(TF) | 1.0 | $27.72 \pm 0.4$ | $49.44 \pm 0.03$ | $29.66 \pm 1.13$ | $20.87 \pm 0.12$ | $61.75 \pm 0.00$ | 37.89 |
| | | *Our Method(Fine-tuning)* | | | | | |
| Qwen3-4B + WeMask(SFT) | 0.1 | $55.01 \pm 0.18$ | $80.96 \pm 0.00$ | $86.69 \pm 0.29$ | $37.42 \pm 0.00$ | $\mathbf{64.54 \pm 0.00}$ | 64.92 |
| Qwen3-4B + WeMask(SFT) | 0.3 | $54.65 \pm 0.23$ | $80.47 \pm 0.00$ | $86.95 \pm 0.15$ | $37.45 \pm 0.00$ | $64.06 \pm 0.00$ | 64.73 |
| Qwen3-4B + WeMask(SFT) | 0.5 | $54.38 \pm 0.03$ | $81.01 \pm 0.00$ | $\mathbf{87.11 \pm 0.44}$ | $\mathbf{37.55 \pm 0.00}$ | $63.76 \pm 0.00$ | 64.76 |
| Qwen3-4B + WeMask(SFT) | 0.7 | $\mathbf{55.02 \pm 0.15}$ | $\mathbf{81.23 \pm 0.00}$ | $87.00 \pm 0.94$ | $37.52 \pm 0.00$ | $63.62 \pm 0.00$ | 64.88 |
| Qwen3-4B + WeMask(SFT) | 1.0 | $52.40 \pm 0.16$ | $77.20 \pm 0.00$ | $85.07 \pm 0.00$ | $34.47 \pm 0.00$ | $62.05 \pm 0.00$ | 62.24 |

an approximately fixed position in the attention similarity space. Since attention scores are computed as inner products, $li0 = q_i^\top k_0$, a directionally invariant key induces stable similarity patterns that vary little with the input. Consequently, such keys act as fixed reference points in self-attention. This interpretation is consistent with prior findings showing that highly similar hidden states will induce rigid representations that reduce input sensitivity and representation diversity (Oh et al., 2025). Moreover, earlier studies demonstrates when representations concentrate along a small number of dominant directions, these directions can dominate representation space, leading to degraded representational quality and reduced effective dimensionality (Ethayarajh, 2019; Timkey & Van Schijndel, 2021).

### 4.2. Proposed Method

Motivated by these limitations, we propose a method named **WeMask** (Weight-guided Masking) that selectively suppresses dominant dimensions in the massive activation, thereby restoring the directional diversity required for effective attention computation without altering the overall transformer structure and incurring no additional computational cost. An overview of the method is shown in Figure 6. Pre-attention RMSNorm preserves direction while amplifying dominant dimensions, reinforcing directional rigidity and reducing attention diversity. Based on this observation, we select dimensions with large RMSNorm weights as candidates for suppression, defined as $\mathcal{S}^{(l)} = \text{TopK}\left(\left|w^{(l)}\right|, , k\right)$, where $w^{(l)}$ is the weight in the layer $l$'s RMSNorm, $k$ denotes the number of selected dimensions determined by the mask rate multiplied by the hidden dimension, and $\mathcal{S}^{(l)}$ represents the selected dimensions. After choosing them, we build a mask as:

$$\mathbf{m}^{(l)} \in \{0,1\}^D, \qquad m_d^{(l)} = \begin{cases} 1, & d \in \mathcal{S}^{(l)}; \\ 0, & \text{otherwise.} \end{cases} \quad (4)$$

Then, we use it to mask corresponding dimension in the input to the attention module, as follows:

$$\tilde{\mathbf{h}}_0^{(l)} = \mathbf{h}_0^{(l)} \odot \left(1 - \mathbf{m}^{(l)}\right), \quad (5)$$

where $h$ means the input hidden state of attention. We insert this module before the attention layer in each subsequent layer, starting from the ME Layer to reduce the rigidity of massive activation's direction and train the model.

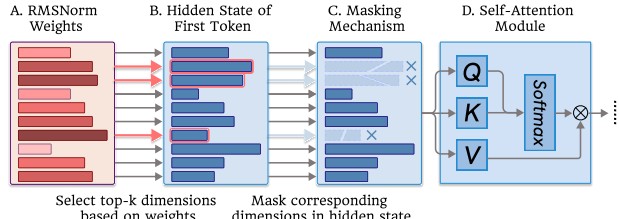

**Layer n (n > ME Layer)**

*Figure 6.* This is the schematic diagram of our methods. We will choose top-k dimensions based on weights then masking the corresponding dimensions in hidden state.

## 5. Experiments

### 5.1. Settings

**Method Details and Training Setups:** We adopt Qwen3-4B as the base model and apply our method both as a training-free inference-time technique and as a training-time strategy across multiple tasks, including instruction fine-tuning, math reasoning, and safety alignment. For each task, we fine-tune the model on the corresponding datasets: FLAN (Wei et al.) and OpenOrca (Lian et al., 2023) for instruction fine-tuning, GSM8K (Cobbe et al., 2021) for math reasoning, and HH-RLHF (Bai et al., 2022) for safety alignment. The context length is set to 4096. Task-specific training configurations, like learning rate and batch size, are provided in the corresponding sections, while all other

*Table 2.* This table presents the performance on math reasoning and safety alignment benchmarks after math-oriented fine-tuning and safety-oriented fine-tuning. TF denotes a training-free inference-time setting without parameter updates, while SFT denotes supervised fine-tuning with parameter updates. **Bold** indicates the best performance under the corresponding experimental settings.

| Method | Mask Rate | GSM8K ↑ | AIME22-24 ↑ | Math500 ↑ | Sorry-Bench↓ | XSTest ↑ |
|---|---|---|---|---|---|---|
| *Our Method(training free)* | | | | | | |
| Qwen3-4B + SFT | - | 20.26 ± 0.43 | 5.92 ± 2.57 | 43.00 ± 1.25 | 3.18 ± 0.13 | 66.22 ± 0.80 |
| Qwen3-4B + SFT + WeMask(TF) | 0.1 | 21.96 ± 0.38 | 6.30 ± 1.28 | 43.60 ± 1.11 | 3.18 ± 0.13 | 67.18 ± 0.93 |
| Qwen3-4B + SFT + WeMask(TF) | 0.3 | 22.01 ± 0.46 | **7.61 ± 2.23** | **43.70 ± 1.73** | 3.48 ± 0.13 | 67.78 ± 0.59 |
| Qwen3-4B + SFT + WeMask(TF) | 0.5 | 21.38 ± 0.17 | 5.93 ± 1.70 | 43.60 ± 1.44 | 3.26 ± 0.13 | 73.78 ± 0.97 |
| Qwen3-4B + SFT + WeMask(TF) | 0.7 | **22.08 ± 0.31** | 7.41 ± 1.28 | 42.87 ± 1.1 | **2.74 ± 0.13** | 74.00 ± 0.59 |
| Qwen3-4B + SFT + WeMask(TF) | 1.0 | 20.89 ± 1.18 | 3.33 ± 1.11 | 38.33 ± 3.21 | 3.04 ± 0.13 | **79.78 ± 0.67** |
| *Our Method(Fine-tuning)* | | | | | | |
| Qwen3-4B + WeMask(SFT) | 0.1 | 21.05 ± 0.88 | 4.07 ± 1.29 | 44.47 ± 0.70 | 2.96 ± 0.34 | 68.82 ± 0.25 |
| Qwen3-4B + WeMask(SFT) | 0.3 | 20.52 ± 0.13 | **8.15 ± 0.64** | **45.40 ± 2.31** | 2.22 ± 0.22 | 69.63 ± 0.34 |
| Qwen3-4B + WeMask(SFT) | 0.5 | 21.91 ± 0.75 | 6.67 ± 2.23 | 41.87 ± 1.72 | 2.74 ± 0.26 | 67.60 ± 1.09 |
| Qwen3-4B + WeMask(SFT) | 0.7 | **22.14 ± 0.53** | 5.92 ± 2.57 | 42.93 ± 1.27 | **1.71 ± 0.13** | **70.59 ± 0.26** |
| Qwen3-4B + WeMask(SFT) | 1.0 | 14.59 ± 0.56 | 1.48 ± 1.70 | 42.73 ± 1.22 | 10.45 ± 0.39 | 69.63 ± 0.51 |

hyper parameters follow the default AdamW settings. In Appendix F, we further use WeMask on Llama-3.1-8B-Instruct and Qwen3-8B, demonstrating our method scales effectively across different model families and parameter sizes.

**Evaluation:** We test 0-shot on several benchmarks, the max new length of output is 512, except GSM8K(128). For every test, we change the random seed and test three times to compute the mean and standard deviation. The benchmark including: MMLU (Hendrycks et al., 2021), PIQA (Bisk et al., 2020), ARC-C (Clark et al., 2018), MathQA (Amini et al., 2019), StrategyQA (Geva et al., 2021), GSM8K (Cobbe et al., 2021), AIME22-24 (AIME, 2024), Math500 (Lightman et al., 2023), SorryBench (Xie et al., 2025) and XSTest (Röttger et al., 2023).

### 5.2. Experimental Results Analysis

**Instruction Fine-tuning.** We first evaluate our method on instruction fine-tuning tasks using Qwen3-4B as the base model, with a global batch size of 256 and a learning rate of 2e-5. Results are reported in Table 1. Qwen3-4B + SFT denotes standard SFT on the training set; Qwen3-4B + SFT+ WeMask (TF) applies our method only at inference time; and Qwen3-4B + WeMask(SFT) jointly fine-tunes the model with our method enabled. The mask rate indicates the proportion of dimensions corresponding to the largest weights that are masked. Our method consistently improves performance across instruction fine-tuning tasks, both in the training-free and fine-tuning settings.

**Math Reasoning and Safety Alignment.** We first apply our method to math reasoning and safety alignment tasks. We adopt Qwen3-4B as the base model, using a global batch size of 64 for math reasoning and 256 for safety alignment, while keeping all other experimental settings identical to those used in instruction fine-tuning. The results are summarized in Table 2. Across both task-specific settings,

incorporating our method consistently improves model performance, indicating that its effectiveness extends beyond instruction fine-tuning. These gains demonstrate that our approach generalizes across different optimization objectives, training paradigms, and data distributions, covering both reasoning-oriented and safety-critical tasks. In particular, on XSTest, standard SFT tends to induce overly conservative refusal behaviors, leading to a noticeable degradation in overall performance. By contrast, integrating our method mitigates this issue by reducing excessive representational rigidity, thereby better balancing safety and helpfulness and substantially restoring overall performance.

**Ablation study.** In Appendix E, we evaluate the effectiveness of our method by comparing it with different masking strategies, including randomly masking a fixed proportion of dimensions and masking the dimensions with the largest activation magnitudes. The results show that these alternative masking methods lead to a substantial degradation in model performance. In contrast, only our method consistently improves performance, demonstrating the effectiveness and necessity of weight-guided dimension masking.

*Table 3.* Performance on safety alignment benchmarks after DPO training. TF and TA denote training-free and training-aware settings, respectively. **Bold** means best performance. Underline means second performance.

| Method | XSTest ↑ | AdvBench ↓ |
|---|---|---|
| Qwen3-4B | 71.93 ± 0.34 | 35.78 ± 0.88 |
| DPO | 72.30 ± 0.51 | 33.80 ± 0.18 |
| DPO + WeMask (TF) | **74.96 ± 0.13** | **32.81 ± 0.44** |
| DPO + WeMask (TA) | 74.74 ± 0.68 | 33.57 ± 0.61 |

**Weight-guided Masking in RL Training.** In this part, we extend our approach to reinforcement learning (RL) and show that it continues to improve the performance of RL-trained models.

For safety alignment, we employ DPO (Rafailov et al., 2023)

to train Qwen3-4B on the HH-RLHF benchmark, randomly sampling 3,000 training instances. The model is trained with a batch size of 8, a maximum sequence length of 1024, and a learning rate of $5 \times 10^{-6}$. Evaluation is performed on XSTest (Röttger et al., 2023) and AdvBench (Zou et al., 2023). For math reasoning, we adopt GRPO (Shao et al., 2024) to train Qwen3-4B on GSM8K, using a batch size of 256, a maximum sequence length of 256, and a learning rate of $1 \times 10^{-6}$. The resulting model is evaluated on AIME 2022–2024 (AIME, 2024) and Math500 (Lightman et al., 2023). As shown in Table 3 and Table 4, our method consistently improves performance across both safety alignment and math reasoning tasks, achieving gains on most evaluation benchmarks. These results demonstrate that our approach generalizes well to reinforcement learning–based training paradigms, highlighting its robustness and scalability beyond supervised fine-tuning.

*Table 4.* Performance on math reasoning after GRPO training. TF and TA respectively denote training-free and training-aware settings. **Bold** is best performance. Underline is second performance.

| Method | AIME22–24 ↑ | Math500 ↑ |
|---|---|---|
| Qwen3-4B | 5.92 ± 0.57 | 43.00 ± 1.25 |
| GRPO | 7.40 ± 3.40 | **43.67 ± 1.92** |
| GRPO + WeMask (TF) | **9.27 ± 1.68** | 43.47 ± 0.12 |
| GRPO + WeMask (TA) | 7.80 ± 1.91 | 43.33 ± 0.12 |

# 6. Discussion: Rethinking Attention Sink from a Representation Perspective

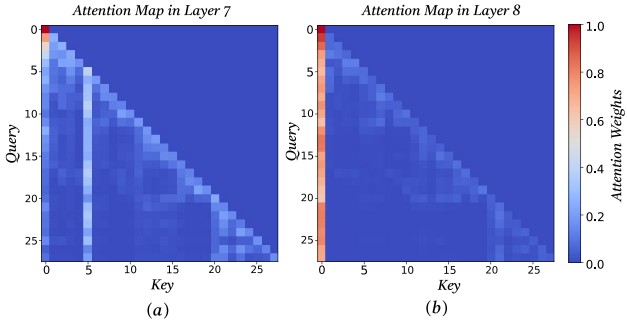

*Figure 7.* (a) shows heatmap of attention weights in the ME Layer (layer 7). (b) shows the layer after ME Layer (layer 8).

Our findings share similarities with prior studies on attention sinks. Previous works, such as Qiu et al. (2025) and Gu et al. (2024), show that attention weights are often heavily concentrated on a single token across multiple heads. This concentration implies a low-rank structure in the attention matrix, reducing the richness of information aggregation. Moreover, attention sinks are observed to persist across different inputs, indicating a degree of input invariance. Similarly, our work focuses on an earlier stage of the model. We find that after the ME Layer the first token's hidden state exhibits an almost input invariant direction while its

magnitude becomes larger than that of other tokens. This behavior suggests a similar low rank effect, but at the level of hidden representations rather than attention weights.

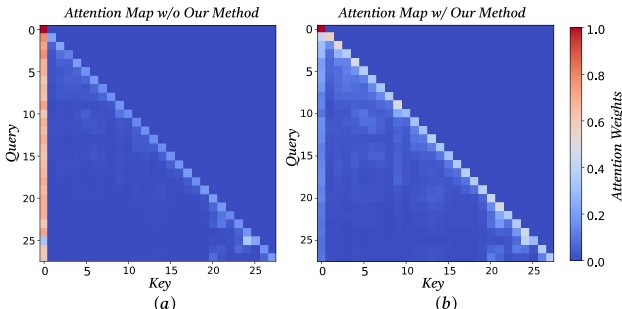

*Figure 8.* (a) shows the attention heatmap without our method. (b) shows the attention heatmap with our method.

Motivated by this connection, we further investigate the relationship between massive activation onset, our proposed intervention, and the emergence of attention sinks. As shown in Figure 7(a,b), attention sinks consistently appear in layers following the onset of massive activation. Notably, the attention sink observed at the ME Layer is not caused by the FFN output of the same layer, as multi-head attention precedes the FFN in the forward pass. Instead, it reflects a directionally rigid representation already consolidated in the residual stream, which becomes explicitly amplified as a massive activation at the ME Layer and subsequently influences attention in later layers. As shown in Figure 8(a,b), our method does not fully eliminate the attention sink but substantially reduces its dominance, resulting in more balanced attention distributions.

Based on these findings, we provide a new perspective on the attention sink phenomenon. We show that attention sinks originate from the ME Layer, where the first token undergoes abrupt magnitude amplification and becomes highly consistent across inputs, collapsing representations into a low-dimensional subspace before entering the attention module. This collapse leads to highly similar keys and queries for the first token, suggesting that **attention sinks are a downstream consequence of massive-activation–induced representation collapse rather than an artifact of the softmax operation**, as emphasized in prior work (Ruscio et al., 2025; Xiao et al., 2024). Importantly, we find that completely eliminating attention sinks is suboptimal: fully removing the sink consistently degrades performance, whereas moderate attenuation preserves useful information while improving overall results. This indicates that attention sinks encode beneficial signals but become harmful when their representations are overly rigid, and that partially relaxing this rigidity yields better model performance.

# 7. Conclusion

In this paper, we analyze the origin of massive activations in large language models and identify the ME Layer as their

point of emergence. We show that once formed, the massive activation token exhibits highly consistent hidden-state patterns across layers, even under diverse inputs, leading to reduced representational diversity and increased directional rigidity. Motivated by this observation, we propose a simple and effective method that relaxes this excessive consistency by intervening directly on hidden-state representations, without modifying the model architecture or training objective. This intervention yields consistent performance improvements across multiple tasks and training settings. Our analysis offers a new perspective on attention sinks, attributing their emergence mitigation to hidden-state dynamics rather than the attention mechanism alone.

## Impact Statement

This paper aims to advance the understanding of internal mechanisms in large language models and to improve their performance through principled representation-level interventions. While enhanced model capabilities may influence downstream applications, we do not identify any ethical concerns or societal risks specific to this work beyond those generally associated with progress in machine learning research.

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

# A. Limitation and Future Works

While our analysis focuses on the emergence and propagation of massive activations in the middle layers, we observe that the final layers exhibit qualitatively different behavior. In particular, the model again produces massive activations in the first token within the last two layers, suggesting that these layers may serve distinct functional roles compared to intermediate layers, such as output consolidation or task-specific representation shaping. However, our current study does not provide a detailed mechanistic explanation for this phenomenon, and a systematic analysis of massive-value formation in the final layers remains beyond the scope of this work.

Moreover, our evaluation primarily considers the post-training setting, where the proposed method is applied after supervised fine-tuning or reinforcement learning. Although we observe consistent performance improvements under this setting, we do not investigate the effects of integrating our method into the pre-training process. Understanding whether suppressing dominant dimensions during large-scale pre-training would lead to similar or even stronger benefits, without adversely affecting representation learning, remains an open and important direction for future research.

# B. Compare the Role of RMSNorm and FFN

As discussed earlier, both RMSNorm and the FFN contribute to the emergence of massive activations. To disentangle their respective roles, we conduct controlled ablation studies by separately removing the RMSNorm preceding the FFN and the FFN itself, and analyze how each modification affects the formation and propagation of massive activations across layers. The result in Figure 9. When the FFN is removed, we observe that the massive-activation token still emerges in the intermediate layers, indicating that earlier components of the network can transiently produce elevated activations. However, these massive activations fail to persist and gradually vanish in deeper layers. This suggests that, without the FFN, the network lacks a mechanism to continuously amplify or maintain such activations as they propagate through the residual stream. In contrast, when the RMSNorm before the FFN is removed, the massive activation remains observable throughout the network. Nevertheless, its magnitude is significantly reduced compared to the original model. This indicates that while RMSNorm substantially influences their scale, likely by reweighting and amplifying specific dimensions of the hidden representation before entering the FFN. Taken together, these results suggest a complementary interplay between the FFN and the preceding RMSNorm in the ME Layer The FFN appears to be the dominant component responsible for generating and sustaining massive activations, whereas the RMSNorm before the FFN plays a crucial role in modulating their magnitude. This interaction helps explain why massive activations emerge sharply and reach extreme values specifically within the ME Layer

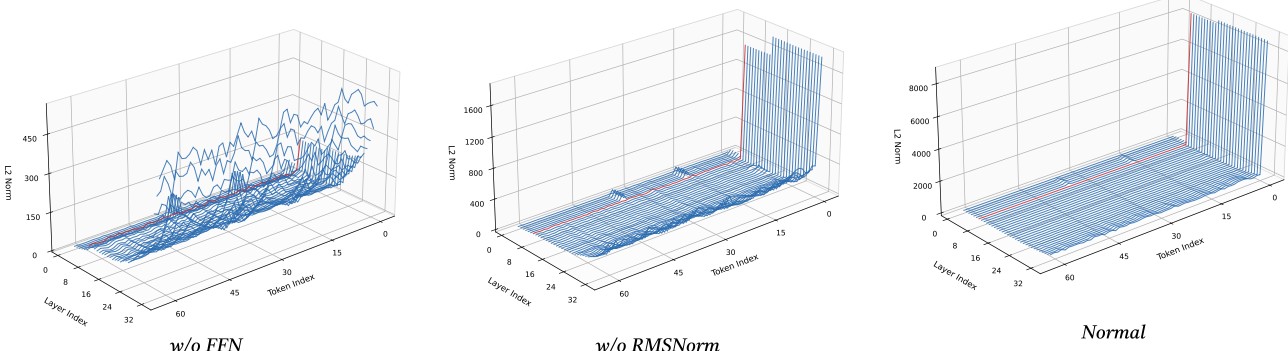

*Figure 9.* The hidden state of the output of DecoderLayer, left figure remove FFN in ME Layer middle figure remove RMSNorm in ME Layer right figure contains all module.

# C. More Experiment Settings

During training, WeMask is applied to every layer following the onset of massive activation. In contrast, during evaluation, we adopt different configurations depending on the task type. For tasks that primarily assess the model's ability to generalize knowledge, we use the same setting as in training and apply WeMask to all layers after massive activation. However, for task-specific evaluations such as mathematical reasoning and safety alignment, WeMask is applied only to the first layer where massive activation emerges during inference.

This design choice is motivated by the different functional roles of WeMask during training and inference, as well as the varying sensitivity of downstream tasks to representational intervention. During training, massive activations emerging after the ME Layer tend to propagate through the residual stream and repeatedly reinforce a directionally rigid representation across subsequent layers. If left unmitigated, this rigidity can accumulate layer by layer, shaping the overall geometry of the hidden-state space. Applying WeMask to all layers following the onset of massive activation therefore acts as a form of representation-level regularization. This encourages the model to learn under reduced directional dominance and to distribute representational capacity more evenly across dimensions throughout the network, leading to more stable and flexible hidden-state dynamics.

During inference, however, the objectives and sensitivities of different tasks diverge. For tasks that primarily assess the model's ability to generalize knowledge across domains or inputs, maintaining consistency between training and evaluation is important. In these settings, we therefore apply WeMask in the same manner as during training, i.e., to all layers following the onset of massive activation. In contrast, task-specific evaluations such as mathematical reasoning and safety alignment rely more heavily on precise intermediate computations and task-specialized circuits formed in deeper layers. Applying WeMask uniformly across all post-ME Layer layers during inference in these tasks may introduce unnecessary interference, potentially suppressing useful task-dependent representations. To address this, we adopt a more targeted intervention strategy: WeMask is applied only at the first layer where massive activation emerges. This setting directly mitigates the initial source of representational rigidity while allowing subsequent layers to operate largely unperturbed, thereby preserving the model's capacity for fine-grained reasoning and decision making. This design balances effectiveness and minimality: WeMask is applied broadly during training to reshape representation learning, while during inference it is selectively deployed to correct the root cause of rigidity without over-constraining downstream computations.

## D. Stability of ME Layer

In this section, we demonstrate that the emergence of the ME Layer is not an incidental phenomenon tied to specific input examples, but a systematic and input-agnostic behavior of the model. We adopt Qwen3-4B as the base model for analysis and evaluate its behavior under a diverse set of input conditions. Specifically, we construct inputs spanning multiple task categories, including commonsense question answering, mathematical problem solving, logical reasoning, and open-ended text continuation. In addition, we vary the input length from short sequences of approximately 10 tokens to long contexts exceeding 1,000 tokens. As shown in Figure 10, regardless of input type or sequence length, Qwen3-4B consistently exhibits massive activation at the same layer, which we identify as the ME Layer This consistency across heterogeneous inputs indicates that the ME Layer reflects an intrinsic property of the model's internal representation dynamics, rather than a task-specific or input-dependent artifact.

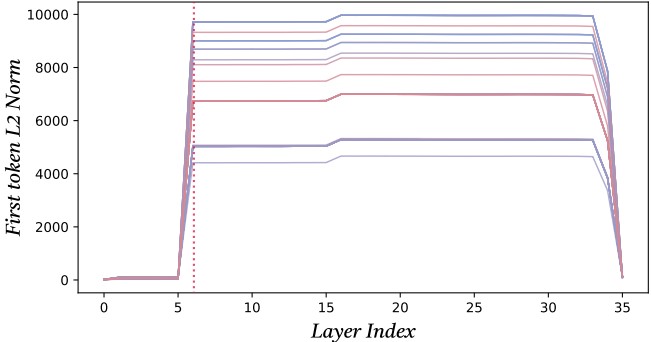

*Figure 10.* L2 norm of the first token across layers for different input instances. Each curve corresponds to a distinct example.

## E. Performance of Different Mask Methods

In this section, we evaluate different masking strategies by incorporating them into the inference stage as training-free interventions, in order to examine their impact on model performance. For each masking method, we adopt the mask ratio that yields the best performance on the corresponding benchmark, as reported in Table 1.

Random Mask randomly masks a fixed proportion of dimensions in the hidden state of the massive-activation token.

*Table 5.* Performance of different masking strategies applied to Qwen3-4B across multiple benchmarks.

| Mask Method | PIQA | ARC-C | StrategyQA |
|---|---|---|---|
| *Qwen3-4B (SFT) as Base Model* | | | |
| Qwen3-4B (SFT) w/o Masking | $80.30 \pm 0.00$ | $86.69 \pm 0.00$ | $65.15 \pm 0.00$ |
| Qwen3-4B (SFT) + Random Mask | $56.02 \pm 1.09$ | $57.52 \pm 1.10$ | $53.67 \pm 0.63$ |
| Qwen3-4B (SFT) + Magnitude Mask | $49.24 \pm 0.00$ | $29.10 \pm 0.00$ | $51.14 \pm 0.00$ |
| Qwen3-4B (SFT) + WeMask | $\mathbf{81.03 \pm 0.22}$ | $\mathbf{87.54 \pm 0.00}$ | $\mathbf{64.63 \pm 0.08}$ |

Magnitude Mask masks the top-$k$ dimensions with the largest activation magnitudes in the massive-activation token. The results are summarized in Table 5. We observe that, except for our method, all alternative masking strategies lead to a substantial degradation in model performance, often causing severe harm to the model's reasoning ability. In contrast, our method consistently improves performance across benchmarks. These results demonstrate that indiscriminately masking dimensions—either randomly or based solely on activation magnitude—destroys critical representational structure, whereas selectively masking dimensions guided by RMSNorm weights provides a principled and effective way to suppress harmful dominance while preserving useful information.

# F. Performance of Other Models

*Table 6.* Using of our method on different models and testing their performance on several benchmarks.

| Model | MMLU(0-shot) | PIQA | ARC-C | OpenbookQA(0-shot) | MathQA(0-shot) |
|---|---|---|---|---|---|
| *Llama3.1-8B-Instruct as base model* | | | | | |
| Llama3.1-8B-Instruct | 46.33 | 75.14 | 74.49 | 76.40 | 24.69 |
| Llama3.1-8B-Instruct + SFT | 48.22 | 84.60 | 79.52 | 80.00 | 28.54 |
| Llama3.1-8B-Instruct + SFT + WeMask(TF) | 48.15 | 84.22 | 79.35 | 79.20 | 28.41 |
| Llama3.1-8B-Instruct + WeMask(SFT) | 47.67 | 84.22 | **79.69** | **81.20** | **30.12** |
| *Qwen3-8B as base model* | | | | | |
| Qwen3-8B | 34.56 | 63.11 | 22.78 | 27.60 | 20.57 |
| Qwen3-8B + SFT | 60.09 | 84.06 | 91.30 | 87.00 | 41.27 |
| Qwen3-8B + SFT + WeMask(TF) | 59.60 | **84.44** | 91.30 | **87.60** | **41.64** |
| Qwen3-8B + WeMask(SFT) | **62.96** | 84.11 | 91.30 | 87.20 | 41.34 |

To evaluate the generality of our method, we further select Llama-3.1-8B-Instruct and Qwen3-8B as base models and fine-tune them using WeMask. We then evaluate the resulting models on MMLU (Hendrycks et al., 2021), PIQA (Bisk et al., 2020), ARC-C (Clark et al., 2018), OpenBookQA (Mihaylov et al., 2018), and MathQA (Amini et al., 2019). The results are reported in Table 6. As shown in the table, compared to the training-free variant, the SFT-based WeMask approach exhibits more stable performance and consistently outperforms the standard SFT baselines across multiple benchmarks. These results demonstrate that WeMask generalizes well across different model architectures and reliably improves model performance.

# G. Compared with Other Methods Which Eliminating Attention Sinks

In the preceding sections, we examined the relationship between our method and the attention sink phenomenon. In this section, we directly compare the effectiveness of our method with existing attention sink removal approaches (Qiu et al., 2025). We adopt the gated attention method to fine-tune the model using supervised fine-tuning (SFT), and evaluate its performance on MMLU (Hendrycks et al., 2021), PIQA (Bisk et al., 2020), ARC-C (Clark et al., 2018), OpenBookQA (Mihaylov et al., 2018), and StrategyQA (Geva et al., 2021). The results are summarized in the table. We observe that, compared to methods that directly suppress attention sinks within the attention module, our approach achieves consistently better performance after fine-tuning. These results further support the validity of our new perspective on attention sinks. Specifically, Qiu et al. (2025) primarily introduces gated modules during the pre-training stage to eliminate attention sinks and improve performance. However, when applied during fine-tuning, such interventions may disrupt representations and inductive biases already learned by the model, leading to suboptimal results. In contrast, our method—applicable in both training-free and fine-tuning settings—provides a simpler and more effective way to improve model performance while mitigating the impact of attention sinks.

*Table 7.* Performance of our method compared to other attention sink removal methods, with the mask rate set to 0.1.

| Model | MMLU(0-shot) | PIQA(0-shot) | ARC-C(0-shot) | OpenbookQA(0-shot) | StrategyQA(0-shot) |
|---|---|---|---|---|---|
| *Qwen3-4B as base model* | | | | | |
| Qwen3-4B | 33.10 | 54.73 | 17.66 | 19.60 | 46.77 |
| Qwen3-4B + SFT | 53.61 | 80.30 | 86.69 | 81.40 | 64.15 |
| Qwen3-4B + Gated Attention | 49.45 | 74.32 | 84.81 | 77.80 | 62.62 |
| Qwen3-4B + SFT + WeMask(TF) | 54.12 | 80.69 | **87.54** | **82.20** | 64.24 |
| Qwen3-4B + WeMask(SFT) | **54.80** | **80.96** | 87.03 | 81.00 | **64.54** |
| *Qwen3-8B as base model* | | | | | |
| Qwen3-8B | 34.56 | 63.11 | 22.78 | 27.60 | 53.32 |
| Qwen3-8B + SFT | 60.09 | 84.06 | 91.30 | 87.00 | 66.77 |
| Qwen3-8B + Gated Attention | 52.66 | 75.41 | 86.95 | 79.40 | 62.53 |
| Qwen3-8B + SFT + WeMask(TF) | 59.60 | **84.44** | 91.30 | **87.60** | **66.99** |
| Qwen3-8B + WeMask(SFT) | **62.96** | 84.11 | 91.30 | 87.20 | 66.69 |

# H. The Universality of ME Layer

In Table 8, we present the ME Layer indices for different models. The results show that the ME Layer is a ubiquitous phenomenon across architectures, and its position is largely consistent within the same model family. For example, both Qwen3-8B and Qwen3-4B-Instruct locate the ME Layer at layer 7.

*Table 8.* The position of ME Layer in different model and their magnification compared to the previous layer.

| Model | Layer | Model | Layer |
|---|---|---|---|
| Phi-3-mini-4k-instruct | 2 | Qwen3-8B | 7 |
| Qwen3-4B-Instruct | 7 | Qwen2.5-7B | 4 |
| Qwen2.5-7B-Instruct | 4 | Qwen2.5-32B-Instruct | 5 |
| Llama3.1-8B | 6 | Llama3.1-8B-Instruct | 7 |
| Mistral-7B-v0.1 | 2 | Deepseek-llm-7b-chat | 2 |

In this section, we will show the L2 Norm of hidden state after RMSNorm, FFN and output in different models to show the universality of ME Layer The Figure 11, Figure 12, Figure 13, Figure 14, Figure 15, Figure 16, Figure 17, Figure 18, Figure 19, Figure 20 shows the output of RMSNorm, FFN and Decoderlayer. We observe that the ME Layer consistently exists across all evaluated models. For models within the same family, such as Qwen3-8B and Qwen3-4B, the ME Layer emerges at the same layer. The output of RMSNorm in Llama3.1 exhibits a different pattern compared to Qwen3. In Llama3.1 or Mistral, the L2 norm of the massive activation token continues to increase after the ME Layer whereas in Qwen3 models it peaks sharply at the ME Layer Despite this difference in post-ME Layer behavior, both architectures share a common characteristic: within the ME Layer the L2 norm of the massive-activation token reaches its maximum, indicating a structurally consistent emergence of massive activations across model families.

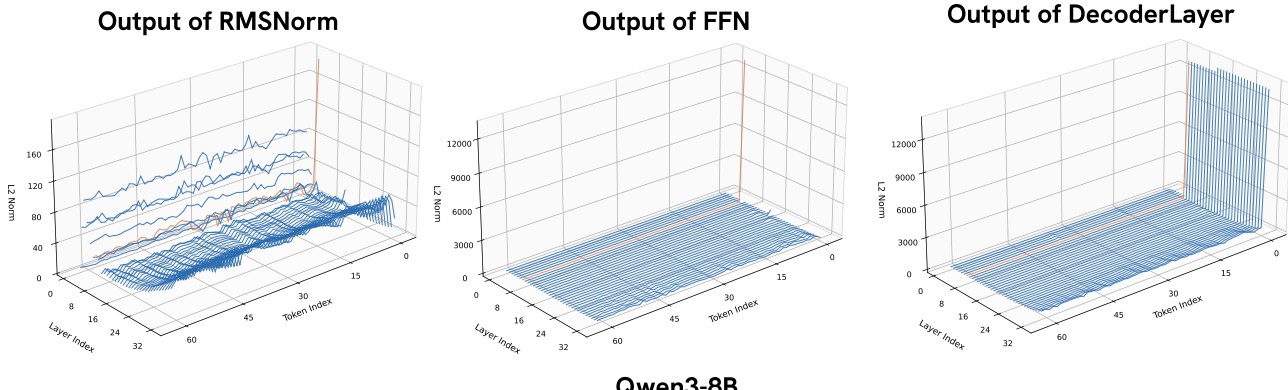

*Figure 11.* The hidden state of the output of RMSNorm, FFN and Decoderlayer on Qwen3-8B

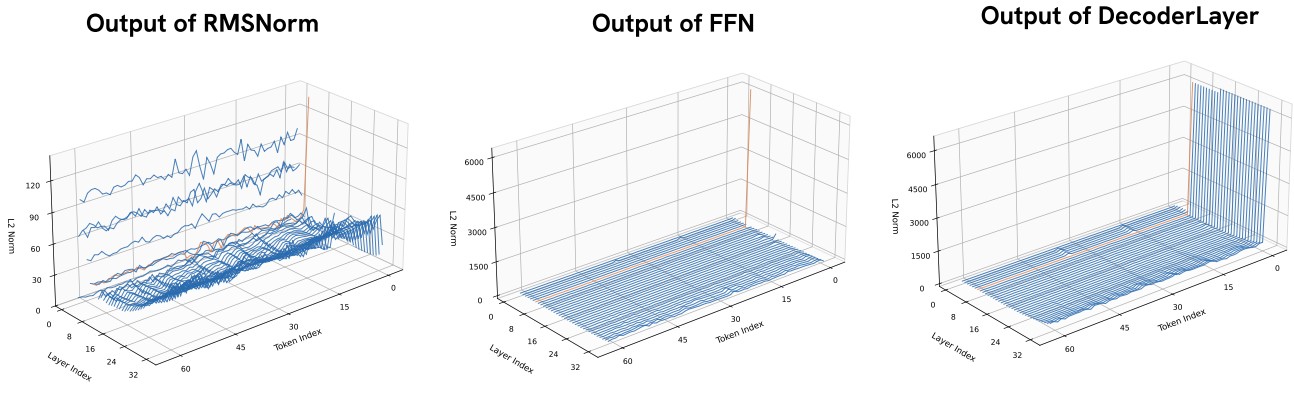

*Figure 12.* The hidden state of the output of RMSNorm, FFN and Decoderlayer on Qwen3-4B-Instruct

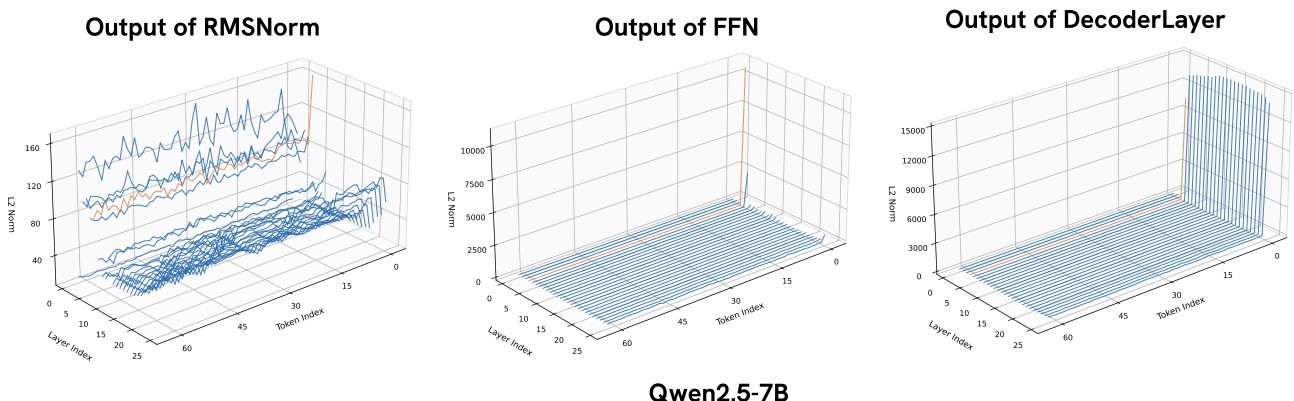

*Figure 13.* The hidden state of the output of RMSNorm, FFN and Decoderlayer on Qwen2.5-7B

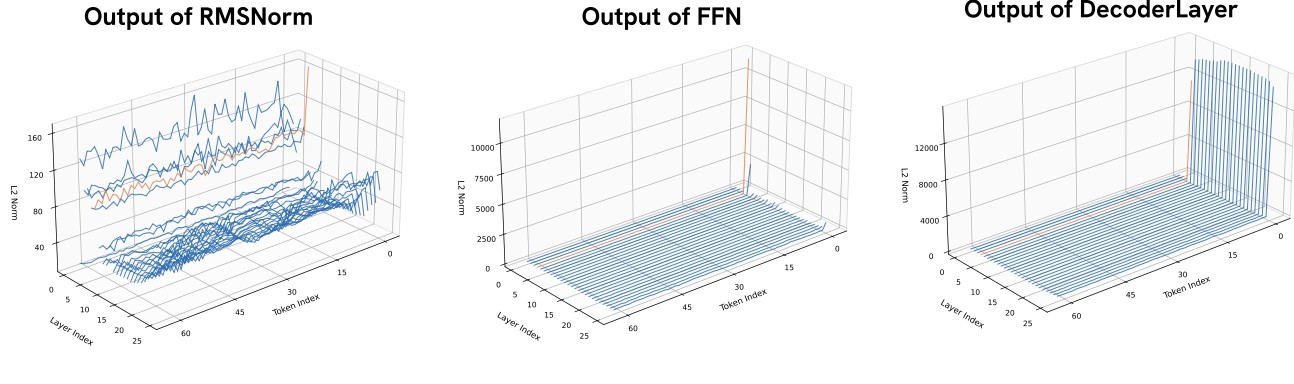

*Figure 14.* The hidden state of the output of RMSNorm, FFN and Decoderlayer on Qwen2.5-7B-Instruct

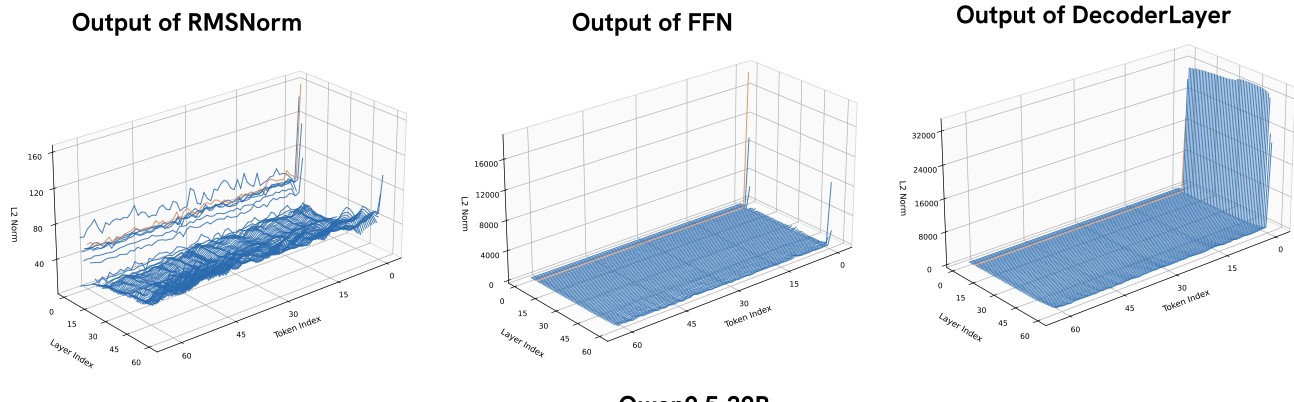

*Figure 15.* The hidden state of the output of RMSNorm, FFN and Decoderlayer on Qwen2.5-32B

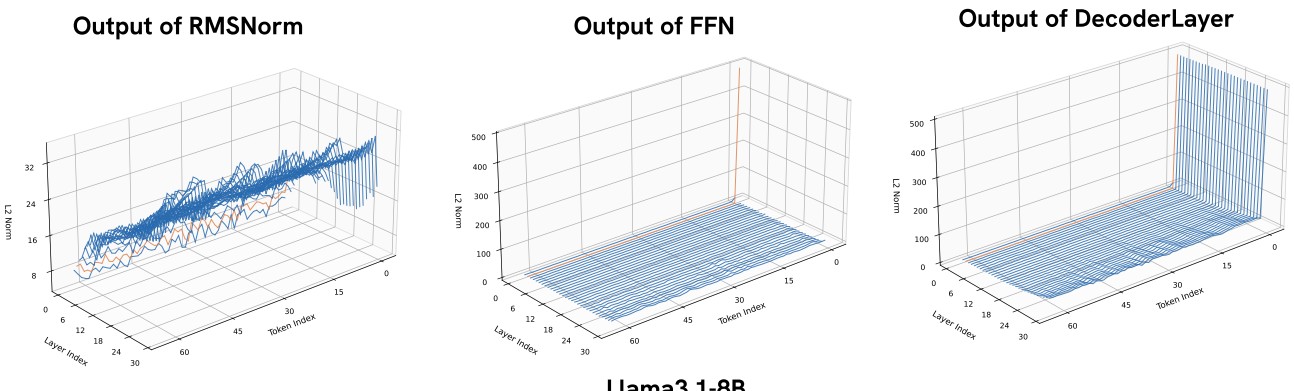

*Figure 16.* The hidden state of the output of RMSNorm, FFN and Decoderlayer on Llama3.1-8B

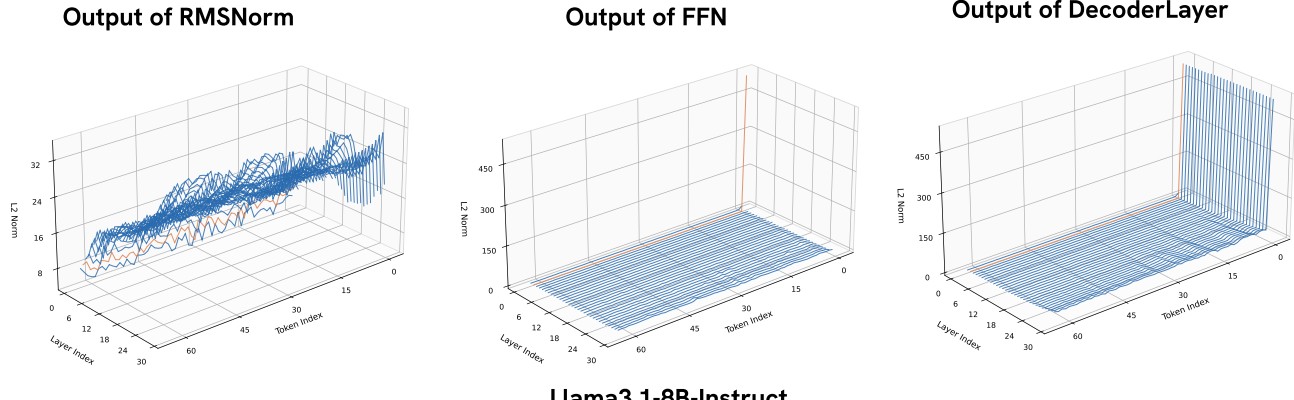

*Figure 17.* The hidden state of the output of RMSNorm, FFN and Decoderlayer on Llama3.1-8B-Instruct

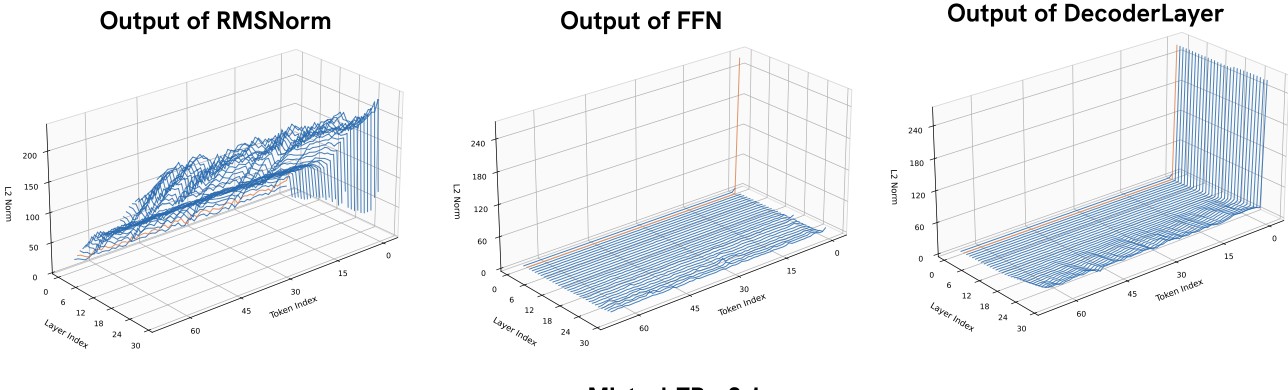

**Mistral-7B-v0.1**

*Figure 18.* The hidden state of the output of RMSNorm, FFN and Decoderlayer on Mistral-7B-v0.1.

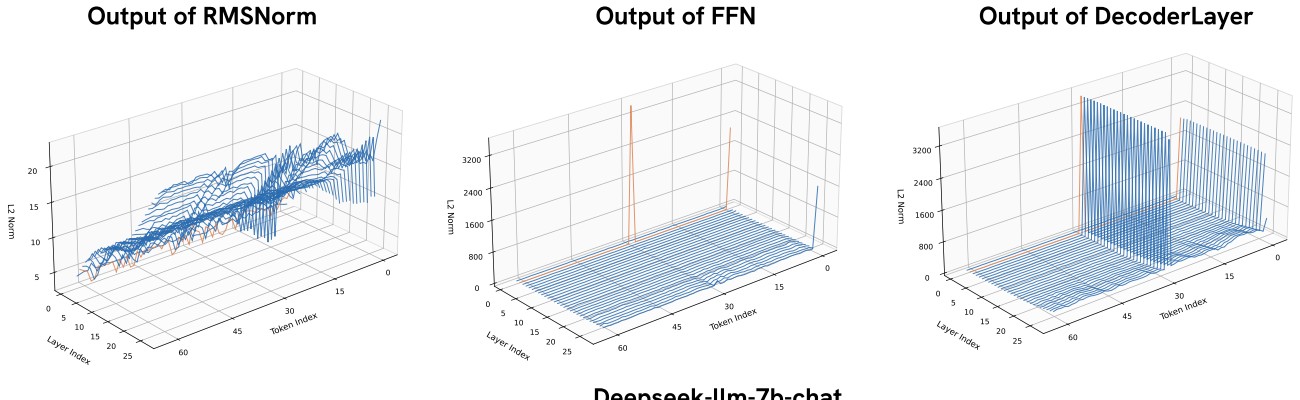

**Deepseek-llm-7b-chat**

*Figure 19.* The hidden state of the output of RMSNorm, FFN and Decoderlayer on DeepSeek-llm-7b-chat.

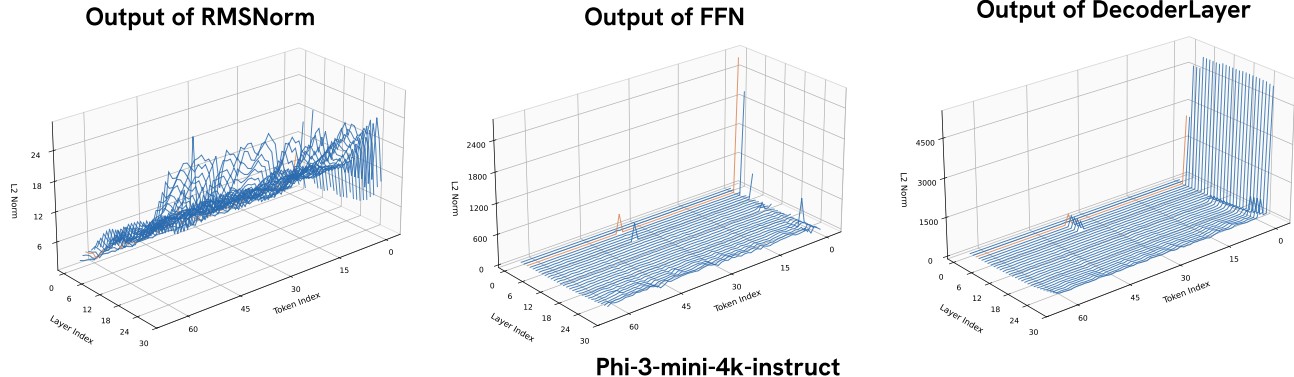

**Phi-3-mini-4k-instruct**

*Figure 20.* The hidden state of the output of RMSNorm, FFN and Decoderlayer on Phi3-mini-4k-instruct.

