# OpenReview forum: "A Single Layer to Explain Them All: Understanding Massive Values in Large Language Models"
_ICML.cc/2026/Conference — ICML 2026 regular_

### Official Review · Reviewer_gV5e · 2026-03-12

**Soundness:** 2
**Presentation:** 3
**Significance:** 2
**Originality:** 1
**Overall Recommendation:** 3
**Confidence:** 4

**Summary:**

The paper studies massive activations in LLMs, where the representations of some tokens get values orders of magnitude larger than the rest. The authors identify a specific layer, the Massive Emergence Layer (ME Layer), where these activations first appear and are kept constant across layers. Within the ME Layer, RMSNorm and the FFN jointly drive the emergence by disproportionately amplifying certain dimensions of the first token. Once formed, the massive activation becomes directionally rigid and nearly input-invariant, reducing representational diversity. To address this, the authors propose WeMask, which selectively masks dimensions corresponding to the largest RMSNorm weights before attention, and report improvements across instruction following, math reasoning, and safety tasks.

**Compliance With Llm Reviewing Policy:**

Affirmed.

**Final Justification:**

Rebuttal Acknowledgement

**Key Questions For Authors:**

- The mask is defined as 1 for dimensions to suppress and 0 elsewhere, then applied as multiplication by (1 − m). Why not simply define the mask as 0 for suppressed dimensions and 1 otherwise, and multiply directly?
- Why are results not reported on the officially instruction-tuned Qwen3 models (i.e., the released post-trained checkpoints) rather than only on base models fine-tuned by the authors? Evaluating WeMask on top of already post-trained models would provide a more realistic assessment of its practical value.

**Limitations:**

yes

**Strengths And Weaknesses:**

Strengths:

- The observation that reducing attention sinks improves performance is a valuable finding.
- The paper is generally well written and easy to follow, with nice figures.


Weaknesses:

- Limited novelty. Most of the analysis in Section 3 has been presented in previous work without properly acknowledging. [Cancedda et al., 2024](https://aclanthology.org/2024.acl-long.263.pdf) already showed for Llama 2 models that early layer FFNs of the BOS token residual stream write in a "dark subspace" that enable attention sinks. And that this is stable across layers, see Figure 9 of [Cancedda et al., 2024](https://aclanthology.org/2024.acl-long.263.pdf) and Figure 14 of [Ferrando et al., 2024](https://aclanthology.org/2024.emnlp-main.965.pdf). The paper should clarify more explicitly what it contributes beyond this prior work.
- The paper does not convincingly motivate why the directional rigidity of massive activations is a problem worth addressing. Attention sinks serve a functional purpose, they let attention heads "do nothing". The paper states that "fully removing the sink consistently degrades performance," which creates a tension, if attention sinks are at least partly beneficial, what is the principled justification for suppressing the dimensions that produce them?
- Most gains fall under 1 point across benchmarks, making it hard to tell whether the differences are meaningful or just noise. Moreover, the optimal mask rate varies by dataset, with no clear way to choose it in advance, which limits the method's practical usefulness.


Typos:
- line 239, 2nd column: "without"
- line 260, 2nd column: direction $\hat{h}_0$
- line 313, 2nd column: missing year in Wei et al citation.

---

> ### Author Rebuttal · Authors · 2026-03-31
>
> We thank reviewer for the questions
>
> >***Limited novelty.***
>
>
> In (arXiv:2402.09221) and (arXiv:2403.00824), they report the BOS token residual stream write in a "dark subspace" and this stability across layers. We want to emphasize that our contribution is fourfold:
> - **Emergence localization**: we identify a specific Massive Emergence Layer (ME Layer) where massive activations arise abruptly, rather than treating them only as a persistent cross-layer property;
> - **Layer-internal mechanism**: we show that this emergence is jointly driven by pre-FFN RMSNorm and FFN, revealing where the amplification comes from;
> - **Hidden-state-level account of attention sinks**: we connect the emergence and propagation of massive activations to downstream attention sinks through increased directional invariance;
> - **Mechanism-driven intervention**: based on this analysis, we propose WeMask, which selectively masks dimensions associated with large RMSNorm weights and yields consistent improvements in training-free, SFT, and RL settings.
>
> We respectfully believe that our novelty lies in providing an **emergence → propagation → downstream sink effect → targeted mitigation** account, rather than merely reporting BOS/first-token stability. We will add a clearer comparison to them in related work.
>
>
>
> >***Attention Sink Removing***
>
> We would like to clarify that **the motivation of our paper is not to remove attention sinks themselves**. Instead, our intervention targets the **massive activations in the hidden state embeddings**, which are highly aligned across inputs and therefore weakly related to input-specific semantics. By masking the dimensions that amplify this representation **before attention**, we reduce the emergence of massive activations, which in turn weakens the attention sink. At the same time, we agree that attention sinks do serve some functional role. This is an important distinction between our work and prior methods that aim to eliminate attention sinks entirely(arXiv:2505.06708). As discussed in **Section 6**, **WeMask does not remove the sink, but only attenuates it**, allowing the model to preserve useful head functionality while making attention more responsive to the semantic content of other tokens.
>
>
>
>
> >***Small gains, possible noise, and dataset-dependent mask-rate selection.***
>
>
> - First, to reduce the possibility that the observed gains are due to randomness, we conducted **multiple runs and report the corresponding variance**. The results show that the improvements are consistent across runs, suggesting that even gains below 1 point are not merely noise.
> - Second, while some gains are modest, they are more substantial on tasks such as **math and safety, with 2–4 point improvements** on AIME, Math500, and XSTest in Table 2. Given that our method is extremely lightweight, we believe that even such stable gains are meaningful.
> - Third, regarding the mask rate, we agree that the optimal value can vary across datasets. However, this is better viewed as a standard **hyperparameter sensitivity issue** rather than a fundamental limitation. Our results show that as long as the mask rate lies within a reasonable range, **WeMask consistently improves performance across different datasets**. Since the method is simple, low cost, and easy to tune on a small validation set, we do not believe this substantially limits its practical usefulness.
>
>
>
> >***Why use 1−m instead of defining the mask directly?***
>
>
> We would like to emphasize that the two definitions are **mathematically equivalent**. This is mainly a matter of notation rather than methodology. We define (m=1) for suppressed dimensions so that the mask directly represents the subset selected for suppression, which aligns naturally with the mask rate. The actual operation, multiplying by (1-m), is therefore equivalent to defining the mask as 0 on suppressed dimensions and 1 elsewhere, and multiplying directly.
>
>
> >***No official Qwen3 post-trained eval***
>
> We evaluate WeMask on Qwen3-4B-Instruct-2507 using PIQA, comparing fine-tuning Qwen3-4B-Instruct vs. fine-tuning the raw Qwen3-4B on our dataset. We find that applying WeMask to Qwen3-4B-Instruct yields performance similar to the version without WeMask, while applying WeMask during fine-tuning from the raw Qwen3-4B leads to better performance.We believe this is because **WeMask is more effective on raw models**: it suppresses the dominant, weakly semantic direction induced by massive activations before it is reshaped by SFT, allowing the model to learn better task-relevant semantic representations during alignment.
>
> ||PIQA|
> |-|-|
> |Qwen3-4B + WeMask(SFT)|81.23|
> |Qwen3-4B-Instruct + SFT|79.54|
> |Qwen3-4B-Instruct + WeMask(SFT)|79.43|
>
> >***Typos***
>
> Thanks reviewer's question, we have aleady fixed these typos in our paper.

---

> > ### Author Rebuttal · Reviewer_gV5e · 2026-04-04
> >
> > While I appreciate the clarification and further results, I am still not convinced that the claimed contributions substantially go beyond prior work. In particular, [Cancedda et al., 2024](https://aclanthology.org/2024.acl-long.263.pdf) already shows that an early-layer FFN (MLP) produces a large-norm vector in a specific subspace that is then propagated across layers and acts as an attention sink, which corresponds closely to the mechanism described here as the Massive Emergence Layer. While the emphasis on RMSNorm and the end-to-end narrative are useful clarifications, they do not, in my view, sufficiently differentiate the work at a conceptual level. Overall, the paper would benefit from positioning itself more clearly as an incremental extension of this existing line of analysis.
> > I will raise my score to weak reject.

---

> > > ### Author Response · Authors · 2026-04-04
> > >
> > > Dear Reviewer gV5e,
> > >
> > > We thank for raising scores and we apology we didn't give a comprehensive comparison in the first round rebuttal because the limitation of characters.
> > > We agree that the paper can better clarify its position with respect to this prior line of work, particularly by emphasizing how our contributions build upon. The extension of our works mainly includes four aspects: (i) single layer localization of emergence, (ii) a joint RMSNorm and FFN mechanism, (iii) a hidden state directional rigidity explanation linking emergence to attention behavior, and (iv) a targeted mitigation method validated across post-training settings. Here is the details,
> > >
> > > - **From a spectral description to a layer-localized mechanism**. [1] explain attention sinks mainly through spectral “dark signals” in the residual stream, and show that some MLP components write into this subspace. In contrast, our work identifies a specific transformer layer, the ME Layer, where massive activations emerge abruptly, and shows that this layer-localized emergence is consistent across model families and sizes. Compared to [1] which only focus on Llama2, we explore more model including Qwen3/Llama3.1/Mistral/Deepseek/Phi4.
> > >
> > > - **Joint attribution to pre-FFN RMSNorm and FFN, rather than MLP writing alone**. A key difference in our analysis is that we do not attribute the phenomenon only to the FFN/MLP. In Section 3.1, we show that massive activation emergence in the ME Layer is jointly driven by the pre-FFN RMSNorm and the FFN: RMSNorm disproportionately amplifies the massive-activation token on high weight dimensions, and the FFN then sharply magnifies that already biased representation. This RMSNorm and FFN interaction is central in our account and is not part of the main mechanistic conclusion in [1], where RMSNorm is absorbed into downstream matrices for convenience rather than analyzed as an explicit causal contributor.
> > >
> > > - **Hidden-state directional invariance influence the emergence to attention behavior.**
> > > Beyond locating where the large activation appears, our paper characterizes what happens after it appears: once formed, the massive activation token becomes highly stable in direction across inputs and across later layers. We then show how this input invariant hidden state direction induces rigid in self attention, thereby reducing contextual diversity and giving a hidden state level explanation for why sink like attention arises. In other words, our explanation to downstream attention behavior is not only “MLP writes a large vector,” but “RMSNorm and FFN create a token whose direction becomes input invariant, and this representational rigidity then leads attention sink.”
> > >
> > > - **A targeted intervention derived from the mechanism**
> > > [1] primarily analyze pretrained LLaMA2 models and study spectral filtering while preserving attention sinking. By contrast, we derive a direct intervention, weight guided masking based on large RMSNorm weight dimensions, that is applied from the ME Layer onward, and we show improvements across training-free inference, SFT, DPO, and GRPO settings, as well as across multiple tasks and model families. So our contribution is not only interpretive but also a mechanism-driven mitigation strategy with broad empirical validation. **[1] explicitly note that their study is limited to pretrained LLaMA2 before instruction/safety fine-tuning, whereas our experiments extend into post-training regimes and downstream performance**
> > >
> > > [1] Spectral Filters, Dark Signals, and Attention Sinks

---

### Official Review · Reviewer_ojKn · 2026-03-12

**Soundness:** 3
**Presentation:** 3
**Significance:** 2
**Originality:** 3
**Overall Recommendation:** 3
**Confidence:** 4

**Summary:**

This paper investigates massive activation tokens in large language models and analyzes how their hidden states behave after the ME layer. The authors find that these hidden states become highly similar across different inputs, which may lead to rigid attention patterns. Based on this observation, they propose a masking strategy WeMask based on RMSNorm weights, and evaluate its effectiveness on several benchmarks.

**Compliance With Llm Reviewing Policy:**

Affirmed.

**Key Questions For Authors:**

Please refer to strengths and weaknesses.

**Limitations:**

Yes

**Strengths And Weaknesses:**

- Soundness: The paper presents a systematic study of massive activation tokens and analyzes how hidden states behave after the ME layer. The experimental evaluation is thorough, and the proposed WeMask method is proved effective on multiple models and benchmarks. However, I feel some experimental details are missing. For example, Section 3 does not clarify the inputs used for the analysis (whether they come from the same task or dataset). Also, the analysis in Section 3 is mainly conducted on the Qwen model, and it would be helpful to check whether the same phenomenon holds for other models. The choice of the top-k candidate dimensions is also unclear, and an ablation over different k values and mask rates would strengthen the analysis.

- Presentation: This paper is generally well structured and easy to follow. The motivation, analysis, and proposed method are presented in a clear order. There might also be a typo: lines 205–206 Figure 4 \rightarrow Figure 3?

- Significance: The problem studied in the paper is interesting, and the connection to attention rigidity and attention sink provides some useful insights. But I feel the link between the analysis and the proposed mitigation could be clearer. Section 3 attributes the issue to hidden-state directionality, while Section 4 addresses it by masking dimensions with large RMSNorm weights, and the connection between these two aspects is not fully explained. Also the evidence in Section 3 mainly shows that hidden states across different inputs become highly similar, but it is unclear whether this necessarily implies the existence of a specific “direction”, rather than a general convergence of representations. In addition, while the proposed method shows improvements on several benchmarks, the performance gains appear relatively modest.

- Originality: This paper identifies an interesting phenomenon and proposes a simple mitigation method that appears to work across several models. It would be helpful if the authors could further justify some of the design choices in the method. For example, Section 4.2 adopts hard masking; I wonder whether softer alternatives, such as scaling, could achieve similar or even better results?

---

> ### Author Rebuttal · Authors · 2026-03-31
>
> Thank you for this helpful suggestion.
>
> >***Soundness.Missing experimental details***
>
> - **In Section 3.2, the analysis is conducted on multiple inputs drawn from different tasks**. The current paper states this in line 234 by "different tasks and input instances". To be more specific, we use different tasks chosen from GSM8K, MMLU and some continuation issues.
> - Regarding model coverage, Section 3 currently uses Qwen3-4B as a case study to analyze the mechanism in detail, because it provides a clean setting for identifying the ME Layer and decomposing the roles of RMSNorm and FFN. However, our claims are not intended to be Qwen specific: **the paper already notes that the ME Layer is observed across model families (Appendix H)**, and **Appendix F further evaluates WeMask on Llama-3.1-8B-Instruct and Qwen3-8B**.
> - In Section 4.2, k is determined by mask rate × hidden dimension which depends on the choose of mask rate which **we have already discussed in Table1 and 2**.
>
>
> >***Presentation***
>
> Thanks for your question, we have fixed these typos in our paper.
>
>
> >***Significance.Weak mechanism–method link***
>
> Thank you for this thoughtful comment.
>
>
> * Our connection is the following: in Section 3, we show that **after the ME Layer, the first-token hidden state becomes highly stable across inputs and layers**, while in Section 4.1 we analyze why this stability persists when entering attention. In particular, pre-attention RMSNorm largely preserves the dominant orientation of the hidden state and further amplifies the already dominant components. Therefore, if the post-ME representation is concentrated on a small set of dominant dimensions, RMSNorm will reinforce precisely those dimensions, causing the resulting key to occupy an approximately fixed position in attention space. This is the motivation for Section 4.2: we mask dimensions with large RMSNorm weights because **these are the dimensions most responsible for preserving and amplifying the rigid dominant orientation identified in Section 3**. In other words, **the masking rule is not ad hoc; it is directly derived from the mechanism analyzed in Sections 3.1 and 4.1.**
>
> * Our claim is not merely based on the fact that hidden states become more similar. Rather, it is based on the combination of three observations: (i) the hidden-state pattern after the ME Layer remains visually similar across layers, (ii) cosine similarity across different inputs remains nearly unchanged after the ME Layer, and (iii) **the representation is highly concentrated on dimensions associated with large RMSNorm scaling factors**. Taken together, these results support the interpretation that **the massive-activation token is dominated by a stable low-dimensional orientation**, rather than only exhibiting generic convergence in norm or overall representation statistics. We will revise the wording to make clear that **our claim is a dominant-direction / low-dimensional concentration interpretation of the evidence**, not an unsupported assertion of a perfectly fixed 1-D direction.
>
> * Regarding the magnitude of the gains, we agree that they are moderate in absolute value on some benchmarks. However, we would emphasize three points. **(1) The method is intentionally lightweight**: it does not modify the transformer architecture and introduces no additional computational module beyond masking selected hidden-state dimensions before attention. **(2) Despite its simplicity, it improves performance consistently across training-free inference, supervised fine-tuning, and RL-based settings**, and remains effective across different task families including instruction following, math reasoning, and safety alignment. **(3) It can mitigate the influence of attention sink with no performance drop.** We believe this consistency is important because **the paper’s main goal is mechanistic understanding with a targeted intervention**, rather than proposing a heavily engineered optimization method aimed solely at maximizing benchmark gains.
>
>
>
>
> >***Originality.Under-justified design choice***
>
> Thanks for youe question, we have tried the soft mask at first and here is the results. We found that the performance of soft mask is worse than hard mask. We think it may because **soft masking only attenuates, but does not sufficiently disrupt**. The other important point is that soft mask also **less effective at reducing the attention sink phenomenon** discussed in Section 6.
>
> | |MMLU (TF)|MMLU (SFT)|PIQA(TF)|PIQA (SFT)|
> |--|--|--|--|--|
> |Scaling factor 0.25|52.85|53.44|80.36|80.74|
> |Scaling factor 0.5|49.84|50.20|80.41|80.58|
> |Scaling factor 0.75|44.61|45.35|80.52|80.74|
> |WeMask| 54.47 | 55.18 |81.20 |81.23|

---

### Official Review · Reviewer_PSaU · 2026-03-12

**Soundness:** 3
**Presentation:** 3
**Significance:** 3
**Originality:** 3
**Overall Recommendation:** 4
**Confidence:** 4

**Summary:**

This paper investigates how massive activations emerge abruptly in a single specific layer (termed the Massive Emergence Layer). Because this layer disproportionately amplifies certain feature dimensions, it leads to directional rigidity in the token representations across subsequent layers. To mitigate this, the authors propose a masking method (WeMask) that selectively masks out these dominant dimensions, which restores representational diversity and allows the attention mechanism to behave more flexibly and adaptively in later layers.

**Compliance With Llm Reviewing Policy:**

Affirmed.

**Final Justification:**

Thanks for the response, I maintain my original score.

**Key Questions For Authors:**

1.	I am curious about the relationship between your masking strategy and standard regularization techniques like dropout. Since randomly masking neurons (dropout) often improves generalization, and masking explicitly larger values naturally exerts a greater influence on the network, could it be that for LLMs, discarding these inherently large values simply acts as a more potent form of regularization? How does this targeted removal of large values compare fundamentally to randomly dropping smaller activation values?
2.	Would you consider incorporating mechanistic interpretability techniques (such as circuit analysis or probing) to further substantiate the universality and specific functional role of the ME Layer, demonstrating that its emergence at a specific depth is a structurally grounded phenomenon rather than a random occurrence?
3.	Will the source code and training/evaluation scripts be open-sourced to ensure reproducibility?
4.	As noted in the limitation section, massive activations reappear in the final few layers of the model. What do you hypothesize is the underlying mathematical intuition or mechanistic reason driving this recurrence at the very end of the network?

**Strengths And Weaknesses:**

Strengths:

1. The paper presents an interesting finding, supported by extensive experiments, demonstrating that massive activations emerge abruptly within a single specific layer (the ME Layer). Intuitively and empirically, the authors effectively demonstrate that this phenomenon leads to directional and representational rigidity in subsequent layers.

2. Based directly on their root-cause analysis, the authors propose a targeted dimension-masking method (WeMask) to mitigate this rigidity. Extensive experiments successfully validate the effectiveness of this approach in improving model performance.

3. The paper addresses a fundamental and universal issue in Large Language Models (LLMs), as the ME Layer is consistently observed across various model architectures and parameter sizes. Consequently, the proposed solution demonstrates strong generalizability, yielding consistent improvements across a wide range of downstream tasks.

Weaknesses:

1. The paper does not evaluate the proposed method or phenomenon on multimodal tasks, such as image recognition. Since VLMs process both text and visual inputs, demonstrating the emergence of massive activations and the effectiveness of WeMask in these models would significantly strengthen the paper. Without this, the claims regarding the broad universality of the phenomenon are somewhat questionable.

2. Does the Top-K  need to be manually tuned for different downstream tasks? The current approach seems to rely on a fixed, task-specific hyperparameter. The paper would benefit from discussing or proposing a more adaptive or dynamic mechanism to determine the optimal masking threshold.

3. I recommend including larger-size LLMs in the experiments. While the findings on 4B and 8B parameter models are interesting, extending the evaluation to larger models  would make the empirical evidence much more convincing and validate whether this phenomenon persists at a larger scale.

4. The performance improvements provided by the proposed method appear relatively marginal on certain benchmarks. Furthermore, the experiments lack comparisons with strong baselines, such as established fine-tuning techniques or existing training-free inference methods that specifically target outlier activations, layer-specific interventions, or dimension masking. Including these comparisons on the same tasks would better demonstrate the method's relative superiority.

---

> ### Author Rebuttal · Authors · 2026-03-31
>
> Thank you for your question.
>
> >***W1. No multimodal validation***
>
> Focusing on LLM is also the standard setting in much of the **prior works on related phenomena such as attention sinks and mechanistic analyses**. (arXiv:2505.06708) and (arXiv:2402.17762) Our goal here is to first establish the phenomenon and mechanism in language only transformers. We acknowledge that VLMs introduce additional components such as vision encoders, projectors, and modality-mixing modules, extending the analysis to them requires a separate controlled study and is an important direction for future work.
>
> >***W2.Top-k tuning unclear***
>
> We clarify that **Top-K is not an independently tuned task-specific hyperparameter**. The masked dimensions are obtained by ranking RMSNorm weights and selecting the top fraction defined by the **mask rate**, so **Top-K is automatically determined once the mask rate is fixed**. Our results further show that performance is stable across a broad moderate range of mask rates, with degradation only under overly aggressive masking. This suggests that the method does **not** depend on delicate task-specific Top-K tuning, but on the general principle of suppressing the most dominant RMSNorm-aligned dimensions.
>
>
> >***W3.Need larger-model evaluation***
>
> We use Qwen3-14B as a larger baseline model to test the effectiveness of WeMask. The ME Layer of Qwen3-14B is Layer 7 and after we add our method on Qwen3-14B, we can see that WeMask can also improve the Qwen3-14B's performance.
>
>
> ||PIQA|
> |-|-|
> |Qwen3-14B + SFT|86.24 +- 0.00|
> |Qwen3-14B + SFT + WeMask(FT)|86.29 +- 0.00|
> |Qwen3-14B-Instruct + WeMask(SFT)|**86.40** +- 0.00|
>
>
> >***W4.Marginal gains, weak baselines***
>
> We clarify that our paper is **not mainly a benchmark-optimization work**, but a **mechanistic study** of massive activations and a lightweight mitigation that reduces representational rigidity and attention sinks while largely preserving the original computation. Thus, moderate but consistent gains are the intended outcome rather than the sole goal. Appendix G already provides a direct comparison with a prior attention-sink elimination method (arXiv:2505.06708), showing that our method **consistently performs better after fine-tuning** on the same benchmarks. Unlike prior work, which mainly targets **pre-training-time sink removal**, our method applies to **already trained models** in both training-free and fine-tuning settings.
>
>
>
> >***Q1.Relation to dropout unclear***
>
>
> WeMask is **not a stronger dropout**, but a **deterministic, mechanism-aware correction**. It suppresses the RMSNorm-aligned dimensions that preserve the rigid dominant direction induced by massive activations after the ME Layer, while largely preserving the rest of the representation. Appendix E shows that the gains do **not** come from masking alone: both **random masking** and **magnitude masking** hurt performance, whereas WeMask improves it. Thus, the key is **which dimensions are suppressed**, not generic removal of activations. The selected dimensions are targeted **not simply because they are large**, but because after RMSNorm they dominate the principal direction responsible for the rigidity identified in Section 3.
>
>
>
>
>
> >***Q2.Need mechanistic evidence***
>
> We would like to clarify that our **current evidence already supports the claim that the ME Layer is not a random occurrence**. In particular, we observe that the emergence of massive activations appears consistently across different inputs and tasks at a stable depth, rather than as an isolated phenomenon tied to a specific example. Moreover, our analysis **goes beyond a purely phenomenological observation by linking this emergence to a reproducible pre-FFN RMSNorm + FFN mechanism**, and by showing how the resulting representation propagates through residual connections and constrains downstream attention. Thus, we do not believe it is necessary to establish the main conclusion of the current paper, that the ME Layer is a mechanistically grounded and consistently recurring emergence point, rather than a random empirical artifact. We have already updated these information in our paper.
>
>
> >***Q3.Open-source reproducibility***
>
> Our authors have decided to release the code after the paper is accepted.
>
> >***Q4.Late-layer recurrence unexplained***
>
> A plausible interpretation is that final layers do not build general contextual features, but instead project hidden states toward a smaller set of directions aligned with the final readout. In residual transformers, such concentrated updates can accumulate rapidly, especially when RMSNorm and FFN keep favoring dominant components, making renewed large-magnitude activations near the output layer plausible even if the exact mechanism differs from that of middle layers.

---

> > ### Author Rebuttal · Reviewer_PSaU · 2026-04-03
> >
> > Thank you for your reply, resolved.

---

### Official Review · Reviewer_mfQ6 · 2026-03-12

**Soundness:** 3
**Presentation:** 2
**Significance:** 2
**Originality:** 2
**Overall Recommendation:** 3
**Confidence:** 4

**Summary:**

This paper investigates the emergence of massive activations and attention sinks, focusing on Qwen3-4B. They find that these occur consistently after the massive emergence layer for distinct samples, and for the first token. Based on these insights, the authors design a technique to mask out the specific directions associated with massive activations, thereby improving flexibility and performance.

**Compliance With Llm Reviewing Policy:**

Affirmed.

**Final Justification:**

I recommend a 3: weak reject. This paper does present interesting analyses of Qwen3 4B in the main text, and their proposed WeMask technique is well motivated and performs consistently. Additionally, in terms of soundness, the rebuttal addressed my initial technical concerns.

However, I continue to have reservations regarding originality, as I find the contribution insufficiently distinct from prior work, despite the authors' helpful clarification of how their analysis is positioned relative to existing methods. Additionally, from a presentation and clarity standpoint, the reliance on a single (relatively small) model in the main text makes it difficult to fully assess the work without incorporating results currently deferred to the appendix, despite the author's proposed revisions, which in my view would have to be substantial. As a result, the paper in its present form feels incomplete, and it is unclear how it would read if the supplemental material were more fully integrated into the main narrative.

Overall, while the paper has clear strengths in motivation and responsiveness to feedback, the limited novelty and presentation concerns prevent me from recommending acceptance.

**Key Questions For Authors:**

These questions are presented in the Weaknesses. I repeat them here:
1. How does the analytical section of your work differentiate itself from “Attention Sinks and Compression Valleys in LLMs are Two Sides of the Same Coin”?
2. How do your findings, both analytical and mask-based, generalize across models?
3. How do the additional suggested ablations to WeMask change its performance (changing activation scale rather than binary masking, and probing when the improvements arise and decline)?

**Limitations:**

In addition to the appreciated limitations mentioned by the authors, the primary limitations are the aforementioned points of lack of novelty and additional masking experiments.

**Strengths And Weaknesses:**

## Strengths
This paper provides a mathematically sound approach to its analysis of massive activations and attention sinks. Additionally, its introduction of WeMask is an interesting and well motivated solution to the limitations that massive activations impose.

## Weaknesses
I will separate the weaknesses into the analytical (Sections 1-3) and masking (Sections 4-6) sections.

### Analytical
In terms of novelty, a substantial majority of the analytical contribution has already been demonstrated by prior work, specifically “Attention Sinks and Compression Valleys in LLMs are Two Sides of the Same Coin” (https://arxiv.org/pdf/2510.06477). That work also found the relationship between massive activations and attention sinks, demonstrating a causal relationship between the occurrence of those two properties for first token across models. Because that prior work is not mentioned and therefore a comparison between it and this work is not provided in the text, it is hard to gauge what is truly novel about the analytical sections.

Furthermore, while the choice to focus on a single model throughout the paper is understandable, because Qwen3-4B is relatively small, the generality of the analytical findings are hard to validate without frequently consulting the appendix. Additionally, much of the findings in the appendix are limited to mostly qualitative analyses (Appendix Section H). It would be more compelling to repeat the analyses from Figures 2-5 for such models and show that in the main paper in some of the figures. The way it is currently presented hides cross-model generality.

### Masking
While showing the performance across various mask rates in Table 1 is important, it may be more pertinent to replace the data within Table 1 with Table 7. This is because demonstrating the improvement of WeMask over prior means to combat attention sinks and massive activations is more fitting for the main text, whereas varying mask rate classifies more as an ablation. Regarding WeMask itself, in Table 1, is the Mask Rate column a percentage or a raw decimal? If it is the latter, as it currently seems to be, it is surprising that there is essentially no variation between a mask rate of 10% of the most activating dimensions and 70%. It would be insightful to probe below 10% and above 70% to better understand where these gains arise and decline, respectively.

Furthermore, another intuitive ablation that could be insightful is scaling the activations rather than fully masking them:

$m^{(l)} \in \set{0, \alpha}^{D}$

The intuition there is that perhaps these directions are indeed important to somehow ground the model, and fully removing them may be detrimental. Have the authors conducted such an experiment?

Finally, in Tables 6 and 7, is there data for the models + WeMask(TF), without SFT at all?

Overall, while the WeMask technique is well motivated, the relatively minor gains on larger models and the limited novelty compared to prior work in the analytical sections motivate my score of 2.

---

> ### Author Rebuttal · Authors · 2026-03-31
>
> Thank you for your question.
>
> >***Noverly Statement***
>
> We will cite it more explicitly in the revised related-work section. However, the analytical focus differs in both scope and granularity. Cancedda et al. (2024) develops a **unified theory** showing that massive activations explain both attention sinks and compression valleys, and uses this to motivate a **Mix–Compress–Refine** view of depth-wise computation. By contrast, our paper focuses on **where and how massive activations are generated inside the model**: we identify a specific ME Layer, show that its emergence is jointly driven by pre-FFN RMSNorm and FFN, and analyze how the resulting large-magnitude token propagates through residual connections. We further show that after emergence, this token becomes highly stable across layers and inputs, yielding an input-invariant dominant direction whose persistence into attention explains downstream rigidity from a hidden-state-level perspective.
>
> >***Generality.***
>
> - Appendix H is meant to provide **cross-model empirical support**, not just a qualitative supplement. The main text presents the mechanism, while Appendix H shows its breadth across model families: Table 8 reports ME Layer positions across all evaluated models, and the layerwise plots reveal the same structural emergence pattern. Together, these results support that the phenomenon is **reproducible rather than model-specific**, so we view this mainly as a **presentation issue, not a missing-analysis issue**.
> - We further **repeated the same analysis on Qwen3-8B and observed a highly similar pattern**. Its ME Layer (Layer 7) shows the same key characteristics: both RMSNorm and MLP exhibit strong amplification at that layer, and after it the massive-activation token again becomes highly aligned in direction across inputs. And we have already updated these analysis in our paper.
>
> >***Mask Rate Analysis***
>
> We will swap Tables 1 and 7 in the revision and clarify that Mask Rate denotes the proportion of selected dimensions, not a raw count. The similar performance from 0.1 to 0.7 suggests saturation rather than contradiction: once enough dominant RMSNorm-aligned dimensions are suppressed, gains stabilize. Performance drops at very small rates because the intervention is too weak, and at very large rates because excessive masking removes useful information.
>
> ||PIQA(TF)|PIQA(SFT)|
> |--|--|--|
> |Mask rate = 0.03|78.78|78.13|
> |Mask rate = 0.05|78.67|78.18|
> |Mask rate = 0.08|78.67|78.13|
> |Mask rate = 0.85|78.78|78.18|
> |WeMask|81.20 |81.23|
>
>
> >***Scaling experiments***
>
>  we have tried the soft mask at first and here is the results. We found that the performance of soft mask is worse than hard mask. We think it may because soft masking only attenuates, but does not sufficiently disrupt. The other important point is that soft mask also **less effective at reducing the attention sink phenomenon** discussed in Section 6.
>
> | |MMLU (TF)|MMLU (SFT)|
> |--|--|--|
> |Scaling factor 0.25|52.85|53.44|
> |Scaling factor 0.5|49.84|50.20|
> |Scaling factor 0.75|44.61|45.35|
> |WeMask| 54.47 | 55.18 |
>
> >***No detrimental impact***
>
> We would like to emphasize that **we do not completely remove these directions from the model’s computation**. As shown in Section 3, they are highly similar across inputs and thus largely unrelated to input-specific semantics. Our intervention only suppresses them before attention, reducing their interference so that the model can focus more effectively on semantic information. **After attention, the corresponding component is reintroduced through the residual connection**, allowing information along these directions to still participate in subsequent computation. Moreover, across SFT, GRPO, DPO, and multiple benchmarks, our method consistently improves performance while also reducing the attention sink phenomenon. This suggests that the intervention is not harmful, but instead helps the model better utilize semantic information by mitigating these semantics-irrelevant dominant directions.
>
> >***More results***
>
> We use choose some benchmarks and show the performance of the three models in Table 6 and Table 7 under models + WeMask(TF). It shows that WeMask will also improve model's performance in the vanilla models especially small models.
>
> ||OpenBookQA|ARC-C|
> |-|-|-|
> |Qwen3-4B|19.60|17.66|
> |Qwen3-4B+TF|21.80|23.04|
> |Qwen3-8B|27.60|22.78|
> |Qwen3-8B+TF|28.00|22.95|
> |Llama3.1-8B-Instruct|76.40|74.49|
> |Llama3.1-8B-Instruct+TF|76.00|74.66|

---

> > ### Author Rebuttal · Reviewer_mfQ6 · 2026-04-03
> >
> > I would like to thank the authors for their detailed response. The rebuttal addresses many of my questions, including the lesser effectiveness of the scaling factor and the mask rate analysis.
> >
> > Regarding generality, I agree with the authors that this is largely a presentation issue rather than a missing-analysis issue. However, I want to emphasize that this presentation choice does have substantive implications: the paper makes claims that suggest a degree of universality, but the supporting evidence is not in the main text. As a result, the central narrative somewhat overstates generality relative to what is directly visible to the reader, which weakens the overall argument. Bringing more of this cross-model evidence, at least to some degree, into the main paper would strengthen the claims' generalizability.
> >
> > Regarding novelty, my concerns are only partially resolved. The authors clarify that their contribution focuses on identifying and analyzing the ME layer and the mechanism of emergence and propagation of massive activations. This is more fine-grained than prior work, and this distinction is reasonable. However, I am still not fully convinced that the analytical contribution is sufficiently differentiated in novelty from existing work (including the paper I cited), given the substantial conceptual overlap. In particular, prior work appears to already identify closely related phenomena and dynamics, even if not framed explicitly in terms of an "ME layer," which makes the incremental novelty here somewhat unclear. Additionally, there is a minor issue in the rebuttal where the cited prior work appears to be misattributed. While not significant on its own, it slightly reduces my confidence that the comparison to related work has been carefully verified.
> >
> > Overall, the rebuttal satisfactorily addresses most of my technical questions. However, my remaining concerns center on limited novelty and relatively incremental empirical gains. I therefore update my score to 3.

---

> > > ### Author Response · Authors · 2026-04-03
> > >
> > > Dear Reviewer mfQ6,
> > >
> > > We sincerely thank the reviewer for raising our score to 3 and for the thoughtful feedback. We fully agree with your comments regarding the narrative, and we have revised the paper accordingly.
> > >
> > > 1. Regarding the comparison with related work, you pointed out a potential misattribution. We apologize for the lack of clarity, which was mainly due to the strict rebuttal length limit preventing a more detailed comparison. Here, we provide a more comprehensive clarification:
> > >
> > > - **Different research focus.** Work [1] aims to provide a unified explanation of attention sinks and compression valleys, focusing on the effects of massive activations. However, it does not give a detailed analyze about how massive activations are generated. In contrast, our work focuses on the origin of massive activations, providing a detailed mechanistic analysis of how they emerge. In this sense, [1] studies what massive activations explain, while we study how massive activations arise.
> > >
> > > - **Different analytical perspective.** Work [1] primarily analyzes correlated trends such as BOS norm, sink rate, and entropy. In contrast, our analysis is grounded in the internal mechanisms of the model, specifically examining the roles of RMSNorm and FFN in producing massive activations.
> > >
> > > - **Methodological contribution.**
> > > Beyond analysis, we propose WeMask, a method designed to mitigate the directional rigidity induced by massive activations. This method reduces their impact on attention and consistently improves performance across different settings, including SFT and RL.
> > >
> > > Overall, we believe there is no novelty overlap between our work and [1]. While [1] focuses on the downstream consequences of massive activations, our work provides a source-level mechanistic understanding.
> > >
> > > 2. We also acknowledge that the phenomenon of massive activations itself is not new, and has been discussed in prior works [2][3][4]. However, our novelty lies in systematically tracing the phenomenon from its origin, explaining how it emerges, how it propagates, how it affects attention (including attention sinks), and proposing a new perspective on mitigation: instead of fully removing such effects, we demonstrate that partial suppression is more effective. We have now included the cited work in our revised paper and expanded the discussion accordingly. If there are additional related works you would like us to address, we would be happy to further clarify.
> > >
> > > Finally, regarding the concern about incrementality, we would like to clarify that we **do not claim SOTA performance**. Our motivation is rooted in identifying a weakness: the directional rigidity induced by massive activations, which can contribute to the formation of attention sinks and lead to the over-dominance of certain tokens in attention weights. Based on this insight. So, we propose a method that not only **mitigates the impact of the rigidity induced by massive activations on attention, but also consistently improves model performance to a certain extent across different tasks and training paradigms**.
> > >
> > > We appreciate your follow up and your willingness to increase the score. We would also like to gently note that the score may not yet have been updated. If our response has fully addressed your concerns, we would greatly appreciate it if you could revisit your evaluation accordingly.
> > >
> > >
> > > [1] https://arxiv.org/pdf/2510.06477
> > >
> > > [2] https://arxiv.org/abs/2402.17762
> > >
> > > [3] https://arxiv.org/abs/2410.01866
> > >
> > > [4] https://arxiv.org/abs/2405.19279

---

### Decision · Program_Chairs · 2026-04-30

**Decision:**

Accept (regular)

**Comment:**

This paper generated meaningful discussion, and reviewers generally agreed that the empirical phenomenon is interesting and that the proposed WeMask intervention is potentially useful. The analysis is reasonably careful, the paper is readable, and the rebuttal helped clarify several technical questions. One reviewer remained positive after rebuttal.
That said, the overall reviewer consensus remains below the acceptance bar. The central concern is originality and positioning. Multiple reviewers argued that the main mechanistic story overlaps substantially with prior work on attention sinks and related analyses, and two reviewers remained unconvinced after rebuttal that the paper establishes a sufficiently distinct conceptual advance rather than a more incremental extension. A second concern is that some of the paper's broader claims about universality are stronger than what is directly supported in the main body, with important evidence delegated to the appendix and some experimental details and ablations not prominent enough. While the rebuttal improved the paper and addressed several narrower questions, it did not fully resolve the core concerns around novelty and scope.